# Precision design of stable genetic circuits carried in highly-insulated *E. coli* genomic landing pads

Yongjin Park[1], Amin Espah Borujeni[1], Thomas E Gorochowski[1,2] (ID), Jonghyeon Shin[1] & Christopher A Voigt[1,2,*] (ID)

## Abstract

Genetic circuits have many applications, from guiding living therapeutics to ordering process in a bioreactor, but to be useful they have to be genetically stable and not hinder the host. Encoding circuits in the genome reduces burden, but this decreases performance and can interfere with native transcription. We have designed genomic landing pads in *Escherichia coli* at high-expression sites, flanked by ultrastrong double terminators. DNA payloads >8 kb are targeted to the landing pads using phage integrases. One landing pad is dedicated to carrying a sensor array, and two are used to carry genetic circuits. NOT/NOR gates based on repressors are optimized for the genome and characterized in the landing pads. These data are used, in conjunction with design automation software (Cello 2.0), to design circuits that perform quantitatively as predicted. These circuits require fourfold less RNA polymerase than when carried on a plasmid and are stable for weeks in a *recA*[+] strain without selection. This approach enables the design of synthetic regulatory networks to guide cells in environments or for applications where plasmid use is infeasible.

**Keywords** gene regulatory network; genetic circuit design automation; genome editing; synthetic biology; systems biology
**Subject Categories** Biotechnology & Synthetic Biology; Methods & Resources
**Mol Syst Biol. (2020) 16: e9584**

## Introduction

Cells use regulatory networks, encoded in their genomes, to determine which genes need to be expressed based on cellular needs or to adapt to the environment (McAdams & Shapiro, 1995). Engineers reconstruct such networks as "genetic circuits" by connecting regulatory proteins to produce a desired computational operation (Elowitz & Leibler, 2000; Atkinson *et al*, 2003; Basu *et al*, 2004; Stricker *et al*, 2008; Wang *et al*, 2011; Moser *et al*, 2012; Din *et al*, 2016; Fernandez-Rodriguez *et al*, 2017). The majority of these circuits

have been characterized using plasmids. One motivation is that circuit optimization requires genetic "tinkering" and making mutations to a plasmid is simpler (Gardner *et al*, 2000; Tabor *et al*, 2009; Fernandez-Rodriguez *et al*, 2017). Another is that plasmids amplify regulator expression, making them easier to connect (Brophy & Voigt, 2014). Finally, they increase the expression of fluorescent reporters used to measure circuit function (Gardner *et al*, 2000; Gorochowski *et al*, 2017). However, carrying a circuit on a plasmid also has disadvantages. They can lead to instability, cell-to-cell heterogeneity, and metabolic burden (Summers & Sherratt, 1984; Chiang & Bremer, 1988; Summers, 1991; Stoebel *et al*, 2008; Kittleson *et al*, 2011; Gyorgy *et al*, 2015; Borkowski *et al*, 2016; Wang *et al*, 2016b). This can lead to evolutionary forces breaking a circuit through plasmid loss or mutagenesis to the plasmid or genome (to reduce the copy number; Mayo *et al*, 2006; Stoebel *et al*, 2008; Klumpp *et al*, 2009; Sleight *et al*, 2010; Chen *et al*, 2013; Sleight & Sauro, 2013; Fernandez-Rodriguez *et al*, 2015; Gyorgy *et al*, 2015; Ceroni *et al*, 2018; Liu *et al*, 2018; Moser *et al*, 2018, 2012). Because of these issues, it is standard practice in industrial biotechnology to introduce recombinant DNA in the genome, particularly if selective pressure from antibiotics is impossible or cost-prohibitive (Singh *et al*, 2011; Isabella *et al*, 2018).

Circuits have been designed for the genome for industrial and biomedical applications. For bio-production, circuits that switch on metabolic pathways after biomass accumulation have been encoded in the genome (Gupta *et al*, 2017). For agricultural applications, sensors have been encoded in the genome to improve stability in soil (Brophy *et al*, 2018). Bacteria engineered to be therapeutics have to be able to function in the human body without antibiotics. To this end, circuits have been integrated into the genome to identify environmental niches, respond to inflammation, induce expression in response to a consumed pharmaceutical, or kill the bacterium (Kong *et al*, 2008; Kotula *et al*, 2014; Danino *et al*, 2015; Mimee *et al*, 2015; Lee *et al*, 2016; Riglar *et al*, 2017; Stirling *et al*, 2017; Isabella *et al*, 2018; Chowdhury *et al*, 2019; Naydich *et al*, 2019). A genome-encoded memory switch was shown to be functional for 6 months in bacteria colonizing the gut of a mouse (Riglar *et al*, 2017). Note that plasmids can also be stabilized for therapeutic using addiction systems (Fedorec *et al*, 2019). For eukaryotes, where plasmids often are not available or unreliable (Davidsohn

1 Synthetic Biology Center, Department of Biological Engineering, Massachusetts Institute of Technology, Cambridge, MA, USA
2 Broad Institute of MIT and Harvard, Cambridge, MA, USA
*Corresponding author. Tel: +1 617 324 4851; E-mail: cavoigt@gmail.com

et al, 2015), it is standard to integrate circuits into the chromosome (Antunes *et al*, 2011; Duportet *et al*, 2014; Roybal *et al*, 2016; Gander *et al*, 2017; Gaidukov *et al*, 2018; Chang *et al*, 2019; Jusiak *et al*, 2019).

These circuits were designed by hand to suit the needs of the application. Design automation software seeks to computationally map a desired circuit function to a DNA sequence (Cai *et al*, 2007; Brophy & Voigt, 2014; Vaidyanathan *et al*, 2015; Nielsen *et al*, 2016; Guiziou *et al*, 2018). Cello does this by having the user specify the desired operation (in Verilog) and the sensors to serve as inputs (Nielsen *et al*, 2016). They specify the organism, genetic location of the circuit, and gate technology by selecting a user constraint file (UCF). Using this information, Cello builds the desired DNA sequence and makes quantitative predictions of the circuit performance in each state. The first UCF (Eco1C1G1T1) is for *Escherichia coli* DH10β, the circuit is carried on a p15a plasmid, and the gate technology is based on NOT/NOR gates using orthogonal TetR-family repressors (Stanton *et al*, 2014). Redesigning a circuit for a new context is a simple as selecting a new UCF.

Cello requires that the gates be transcriptional; in other words, their inputs and outputs have to be promoters. To make a UCF, the gate response functions (how the output changes as a function of the input) have to be measured at the same site where the circuit will be carried. Characterizing the response functions in relative promoter units (RPUs) simplifies the prediction of how they can be connected in series to build a circuit (Kelly *et al*, 2009; Nielsen *et al*, 2016). Design automation is only as successful as the quality of the gates; they must produce the same response in the context of different circuits. Achieving this has required insulating the gates through the use of strong terminators, long promoters, and ribozymes (Davis *et al*, 2011; Lou *et al*, 2012; Cambray *et al*, 2013; Chen *et al*, 2013; Mutalik *et al*, 2013; Brophy & Voigt, 2014; Nielsen *et al*, 2016). Carrying a circuit in the genome introduces additional modes of potential interference, including RNA polymerases (RNAPs) from neighboring regions and changes to the macrostructure (Guo & Adhya, 2007; Vora *et al*, 2009; Lasa *et al*, 2011; Mitschke *et al*, 2011; Chong *et al*, 2014; Lybecker *et al*, 2014; Wade & Grainger, 2014; Brophy & Voigt, 2016; Dorman & Dorman, 2016; Yeung *et al*, 2017). In addition, different regions of the genome vary in effective copy number, with the highest being near the origin, so a circuit encoded at a locus may behave differently at another (Chandler & Pritchard, 1975; Schmid & Roth, 1987; Sousa *et al*, 1997; Rocha, 2008; Block *et al*, 2012; Slager & Veening, 2016). Collectively, these effects can cause the expression of a recombinant gene to vary by up to 300-fold depending on where in the genome it is encoded (Bryant *et al*, 2014; Scholz *et al*, 2019).

This work describes the reliable design of genetic circuits for highly insulated "landing pads" in the *E. coli* MG1655 genome. The landing pad positions were identified using random transposon mutagenesis to identify high-expression positions and then confirming that native gene expression is not impacted. The landing pads contain orthogonal phage integration sites so that they can be independently targeted with high efficiency (Datsenko & Wanner, 2000; Choi & Schweizer, 2006; Sharan *et al*, 2009; Wang *et al*, 2009, 2016a; Kuhlman & Cox, 2010; Lambowitz & Zimmerly, 2011; Enyeart *et al*, 2013; Esvelt & Wang, 2013; Santos *et al*, 2013; St-Pierre *et al*, 2013; Gu *et al*, 2015; Jiang *et al*, 2015; Pyne *et al*, 2015;

Bassalo *et al*, 2016). The landing pads are flanked with new ultra-strong double terminators to block transcription into or out of the sites. A NOR gate architecture is developed for the genome, and the response functions for 6 TetR-family repressors are characterized in a landing pad. These data are used to build a UCF (Eco2C1G3T1) for Cello. Design automation is used to create genome-encoded circuits, and they are found to function quantitatively as predicted. The division of regulatory sensors and circuitry across defined positions in the genome represents a step toward the organized design of synthetic regulatory networks for genome-scale engineering projects.

## Results

### Genetic landing pad construction and characterization

Natural terminators are often weak, and pervasive transcription is common in both the sense and antisense directions of the genome (Lasa *et al*, 2011; Cambray *et al*, 2013; Chen *et al*, 2013; Lybecker *et al*, 2014; Wade & Grainger, 2014). If allowed to enter the circuit DNA, genomic RNAP flux can cause the circuit to malfunction (Lee *et al*, 2016). The opposite can also be problematic, where the RNAP flux from the circuit can exit and cause genes encoded in the genome to be expressed incorrectly. This effect is worsened by the fact that synthetic circuits typically produce large swings in RNAP flux. To address these issues, we designed very strong terminators to flank the landing pads and block transcription both into and out of the genetic circuit.

The terminator strength ($T_S$) is a metric that captures how much gene expression changes before and after the terminator (see Materials and Methods for definition; Chen *et al*, 2013). We previously characterized a large library of individual terminators, the majority of which are strong with an average $T_S = 30$ (97% of transcription is blocked). Some terminators block RNAP flux in both directions (bidirectional), but the average strength in the weak direction is low ($T_S = 6$ or 17% transcriptional readthrough). To better insulate the landing pads, a stronger set of terminators was designed by concatenating two terminators in series (Appendix Table S1, Appendix Fig S1). The single terminators were sourced from large libraries of characterized parts (Cambray *et al*, 2013; Chen *et al*, 2013), the terminator prediction software RNIE (Gardner *et al*, 2011), and *E. coli* MG1655 transcriptome data (Materials and Methods). This led to a set of 93 double terminators, including variations in spacers and ordering. Then, the strength of each double terminator was measured using an assay based on placing the terminator between red and green fluorescent protein genes downstream of an inducible promoter (Materials and Methods; Appendix Fig S14; Chen *et al*, 2013). Seventeen were selected that have $T_S = 105$ to 4744 and sufficiently diverse sequences to avoid homologous recombination (Appendix Table S1). Note that the double terminators are relatively small (< 170 bp). Six bidirectional double terminators ($T_S > 105$ forward and $T_S > 25$ reverse) were selected to flank the landing pads (Fig 1A).

The next step was to identify regions of the genome where a landing pad could achieve high levels of expression and not impact growth. Following an approach used by Freddolino and co-workers (Scholz *et al*, 2019), Tn5 transposon mutagenesis was used to

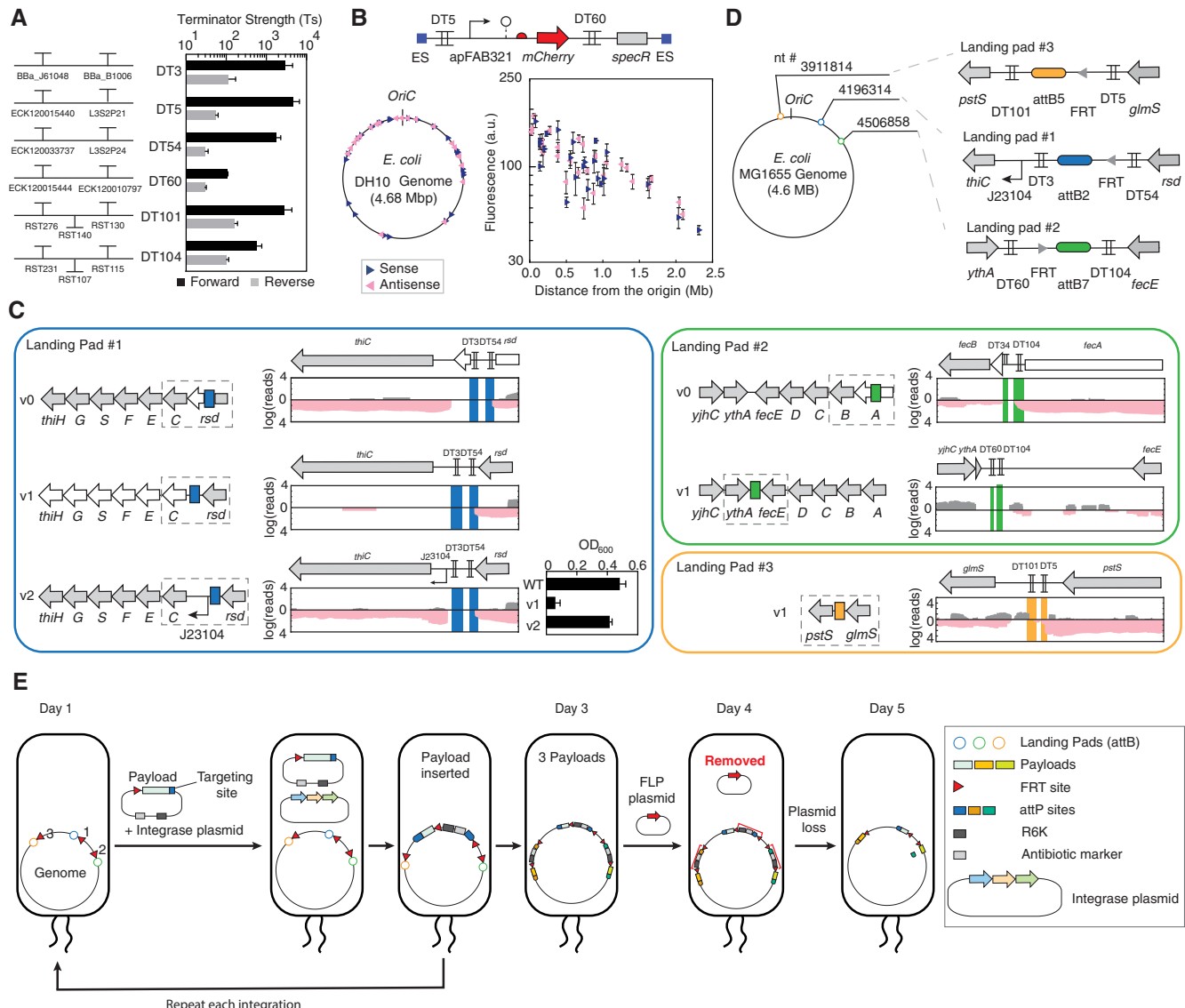

**Figure 1. Design of insulated genetic landing pads.**

A  The six bidirectional double terminators are shown. The terminator strength is shown, calculated as described in the Materials and Methods. Terminator sequences are provided in Appendix Table S1. The means of three experiments performed on different days are shown, and the error bars are the standard deviation of these measurements.

B  The transposon library screen in *E. coli* DH10β. The construct shown was randomly integrated into the genome (ES, end sequences). Sense and antisense insertions are denoted by the navy right and pink left triangles, respectively. Detail information (location, expression levels, and $OD_{600}$) of the insertion locations is provided in Appendix Table S2. The means of three experiments performed on different days are shown, and the error bars are the standard deviation of these measurements.

C  The genomic impact of the landing pad locations. RNA-seq was performed; sense and antisense transcripts are shown in gray and pink, respectively. The dashed squares show the regions of the genome shown in the transcriptional profiles. Genes colored white are those for which we observed large changes upon the insertion of the landing pads. The growth ($OD_{600}$) of *E. coli* MG1655 strains harboring Landing Pad #1 v1 and v2 in Thamine-free medium is shown. The means of three experiments performed on different days are shown, and the error bars are the standard deviation of these measurements.

D  Final selection of three Landing Pads.

E  Schematic showing the steps and time required to insert multiple payloads into the genome. A detailed protocol and the result of integration are provided in Appendix Note S1.

randomly insert a constitutive promoter driving mCherry expression in the genome (Fig 1B). To avoid the high expression being due to transcriptional activity from the genome, the expression cassette was flanked by a pair of bidirectional double terminators. *E. coli* DH10β was selected as a recipient strain due to its enhanced tolerance for foreign DNA and high-throughput transposon library construction (Grant *et al*, 1990; Durfee *et al*, 2008). The transposon was randomly integrated into 50 sites, and the expression and growth impact were measured (Appendix Table S2). As observed previously, the highest expression occurs close to the origin, but

there was significant site-to-site variation even in this region (Fig 1b; Schmid & Roth, 1987; Rocha, 2008; Block *et al*, 2012; Bryant *et al*, 2014; Scholz *et al*, 2019). The directionality of the construct did not systematically impact expression. Three sites were initially selected for the landing pads that had high expression and no impact on growth. In addition, we sought sites that did not disrupt an essential gene or for which we identified putative regions within 10 kb where the landing pad could be moved to avoid impacting host gene expression.

After identifying the landing pads in *E. coli* DH10β, they were moved to *E. coli* MG1655 K-12 because it is more commonly used in industry (Xie *et al*, 2003; Sezonov *et al*, 2007). This was simplified by the observation that the relative expression levels between sites are similar when compared between *E. coli* DH10β and *E. coli* MG1655 K-12 (Appendix Fig S2). Three landing pads were designed based on orthogonal *att* sites so that different phage integrases could be used to direct recombinant DNA to a landing pad (Yang *et al*, 2014). To avoid off-target effects, we selected *att* sites that do not share sequence identity with the *E. coli* MG1655 genome (Materials and Methods). Landing Pads #1, #2, and #3 are based on *att* sites specific to Int2, Int7, and Int5, respectively. Finally, FRT sites were added to remove the antibiotic selection markers from each landing pad. After inserting the landing pads using lambda red recombineering, they are transferred to a clean genomic background using P1 transduction (Materials and Methods).

Landing Pad #1 (v0) was initially located within *rsd*, a repressor of sigma factor D involved in the gene expression in stationary phase (Fig 1D; Jishage & Ishihama, 1999). We were concerned about disrupting this gene, so the landing pad was moved downstream of the gene with what we thought would be enough room to avoid impacting the promoter of the next gene (*thiC*) (v1). However, this was found to completely abolish transcription of the *thi* operon, which is responsible for producing thiamine (Vander Horn *et al*, 1993; Xi *et al*, 2001; Leonardi *et al*, 2003). Indeed, while there was no growth defect in M9 supplemented with 0.4% glucose, casaminoacids, and thiamine, there was a severe growth defect when grown in media lacking thiamine (Fig 1C). Noting that in the wild-type genome this operon is constitutively expressed even when grown in media containing thiamine, we selected a medium-strength constitutive promoter (BBa_J23104; Kelly *et al*, 2009) to be placed downstream of the landing pad. This was found to recover the transcription of the *thi* operon and overcome the growth defect (v2, Fig 1D). The movement of this landing pad and the insertion of the synthetic promoter did not impact the expression of gene expression level from this site (Appendix Fig S3).

Similar experiments were performed to analyze Landing Pads #2 and #3. Landing Pad #2 initially disrupted the first gene of the *fec* operon (Enz *et al*, 1995, 2003). Therefore, we moved it 3.9 kb downstream *fecE* and before the terminator after the *ythA* gene, which is oriented in the opposite direction (Fig 1D). This was found to not impact growth nor transcript levels, while yielding the same level of expression (Appendix Fig S3). The initial site for Landing Pad #3 was found to be only 6.7 kb apart from the well-established Tn7 transposase integrase site (Choi & Schweizer, 2006) that showed no impact on growth and had high level of gene expression (Segall-Shapiro *et al*, 2018). Note that the mCherry constitutive expression cassette yields very similar expression levels across all three sites (< 10% difference), making them interchangeable in carrying constructs. When empty, all three landing pads are small (407, 386, and 901 bp) and show very strong insulation in both the sense and antisense directions (Fig 1D). The final *E. coli* MG1655 strain (YJP_MKC173) contains the three landing pads and no selective markers.

## Genome engineering methodology

Our objective was to simplify the process of genome engineering so that it approaches the ease of plasmid manipulations (Fig 1E). The DNA payloads are cloned into three plasmids (plYJP064, plYJP066, and plYJP070), each containing an *att* site directing it to a landing pad, an antibiotic marker and a single FRT site. A single plasmid (plYJP053) constitutively expresses all three integrases. The payload and integrase plasmids all contain the R6K origin that can be amplified in a $pir^+$ strain, but cannot replicate in *E. coli* MG1655 (Shafferman *et al*, 1982; Metcalf *et al*, 1994). Electroporation is typically performed for the transformation steps, but with the addition of the origin of transfer (OriT) (Fu *et al*, 1991), the plasmids can also be delivered via conjugation by mixing a recipient strain and two donor strains (Materials and Methods). Each payload plasmid has a different antibiotic resistance, which can be removed in a single step by transforming a plasmid containing the FLP recombination, which can be subsequently cured. We attempted to co-transform multiple plasmids to integrate into different landing pads simultaneously, but this approach was unsuccessful. Appendix Note S1 contains a detailed protocol for performing serial integrations.

## RPU reference promoter in the genome

Cello designs circuits by connecting transcriptional sensors and circuits, whose signal carrier is defined as RNAP flux (Canton *et al*, 2008). In previous works we selected a plasmid-borne constitutive promoter (BBa_J23101) for this purpose, which had been identified by others as a good reference promoter to obtain more reliable promoter measurements across laboratories (Canton *et al*, 2008; Kelly *et al*, 2009; Nielsen *et al*, 2016). We moved this promoter to the genome (Landing Pad #1) with a stronger RBS (BBa_B0034) and measured its fluorescence (Fig 2A). This value is used to normalize the promoters of the sensors and gates such that they can be reported in relative promoter units (RPU$_G$; Fig 2B). Note that correcting for the RBS strength and reporting RPU on a per DNA basis would result in the same value as that measured for the plasmid-borne RPU reference promoter. The value of RPU$_G$ in absolute units is estimated to be 0.067 RNAP/s (Materials and Methods).

## Sensor array in Landing Pad #3

To modularize the organization of synthetic regulatory networks, we assigned Landing Pad #3 to carry the sensors and the circuit to be divided between Landing Pads #1 and #2. The sensor array is composed of a set of 7 small molecular sensors. These sensors were selected from a larger set of 12 developed for the "Marionette" *E. coli* strains (Moon *et al*, 2012; Stanton *et al*, 2014; Meyer *et al*, 2019), with several removed because their repressors appear in our

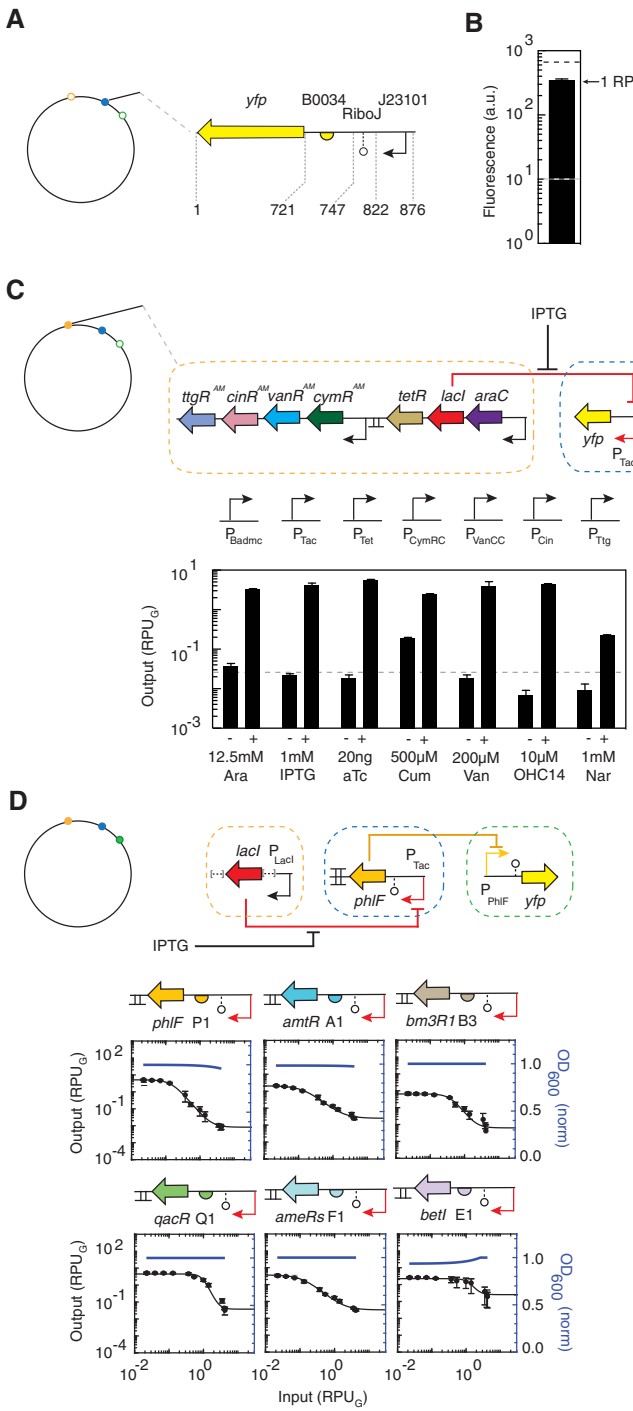

**Figure 2. Characterization of genetic sensors and gates in the genome.**
The genomic landing pads are shown as open circles when empty and full when carrying a genetic design.

A The reference promoter and construct used for the calculation of $RPU_G$ are shown.

B The strength of the $RPU_G$ reference promoter is shown. Gray dotted line shows autofluorescence level, and the black dotted line shows the RPU reference characterized on a p15a plasmid (Nielsen *et al*, 2016; Shin *et al*, 2020). The means of three experiments performed on different days are shown, and the error bars are the standard deviation of these measurements.

C The sensor array is shown; genetic parts are provided in Appendix Table S5. The output promoters associated with each sensor is shown at the top of the bar graph. The detailed characterization of the sensors, including ON/OFF values and response functions, is shown in Appendix Fig S5 and Appendix Table S3. The means of three experiments performed on different days are shown, and the error bars are the standard deviation of these measurements.

D Characterization of NOT gates. The [...] in the construct represent portions of the sensor array not including in the schematic for clarity. Response functions were measured for the following 12 concentrations of IPTG: 0, 2, 5, 10, 20, 40, 60, 80, 100, 200, 400, and 500 μM. The *x*-axis was converted to $RPU_G$ by separately measuring the activity of the IPTG-inducible promoter for these inducer concentrations (Appendix Fig S12). Each response function was fit to the Hill equation (Materials and Methods), and the fit parameters are provided in Table 1. The means of three experiments performed on different days are shown, and the error bars are the standard deviation of these measurements. The blue lines show the impact on cell growth ($OD_{600}$) when the input promoter to the gate is turned on (Materials and Methods). The smooth line is a linear regression to data from twelve inducer concentrations repeated over 3 days (Appendix Fig S8).

## NOT gates in Landing Pads #1 and #2

The NOT gate response functions need to be measured in the same genetic context used to carry the circuits. The genome-encoded IPTG-inducible system in Landing Pad #3 is used as the input ($P_{Tac}$), and the gate is carried in Landing Pad #1 (Fig 2d). The corresponding output promoter driving the expression of YFP is carried in Landing Pad #2. The response function is measured by changing the concentration of IPTG and measuring the output at steady state using flow cytometry (Materials and Methods). However, this results in a function whose *y*-axis is in units of inducer concentration. To change the units to the activity of the input promoter, separate measurements are made for the induction of $P_{Tac}$ in Landing Pad #1 (Appendix Fig S12).

Initially, we characterized a set of 10 gates based on orthogonal TetR-family repressors that had been previously designed to build circuits on a p15a plasmid (SrpR, PhlF, QacR, AmtR, LitR, BM3R1, PsrA, AmeR, and BetI; Stanton *et al*, 2014; Nielsen *et al*, 2016). From the initial set of repressors, we removed SprR, LitR, and PsrA because of crosstalk or toxicity. Several changes were made to the remaining gates to improve their dynamic ranges when carried in the genome: Alternative RBSs were selected for PhlF/BetI, and mutations were made to AmeR to improve binding to its operator (AmeRs) (Appendix Fig S7). To better insulate the gates, we replaced their single terminators with double terminators not already used as part of the landing pads (Appendix Fig s S6 and S14, Appendix Table S4). The response functions of the six gates are shown in Fig 2D (Table 1). The impact of each gate on cell growth is minimal (Fig 2D, Appendix Fig S8; Shin *et al*, 2020).

gate library. Note that this work places the sensor array in a different genomic location and we made several genetic changes to the sensors to optimize their dynamic range in this context (Materials and Methods). The seven regulatory genes are organized as two constitutively transcribed operons (Fig 2C). The output promoter for each sensor was used to drive YFP expression from Landing Pad #1. These sensors produce a 12- to 640-fold induction (Fig 2C, Appendix Fig S5, Appendix Table S3) with low off states and no evidence of crosstalk (Fig 2c and Appendix Fig S5).

## NOR gate design for the genome

To build circuits, the NOT gates have to be convertible to multi-input NOR gates. On plasmids, we found that it was efficient to place two promoters in tandem upstream of the repressor gene (Tamsir *et al*, 2011). Our first genome-encoded gates were based on this compact design, but we found that they did not function well (Fig 3D and F, and Appendix Fig S9). Indeed, this design has several problems that are exacerbated when carried on the genome. The first is that the upstream promoter is inhibited by the binding of repressor to the downstream promoter ("roadblocking"), which is more problematic at lower copy number (Nielsen *et al*, 2016; Shin *et al*, 2020). In addition, maximum repressor expression is less than the sum of the input promoter activities.

A new NOR gate design was developed that splits the gate such that there are two copies of the repressor gene, each of which is driven by a different input promoter (Fig 3B). To avoid homologous recombination, the two genes are encoded in different landing pads. Each gene contains the same ribozyme, RBS, codon usage, and terminator in order to produce the same repressor expression levels in response to a given input promoter activity. Two versions of the PhlF repressor NOR gate were constructed based on the old (tandem) and new (split) gate designs (Fig 3A and B). Both gates were evaluated for 64 combinations of inducers (aTc and ara), and the activity of the output promoters was compared (Fig 3C). The gates perform comparably, noting that the maximum repression is higher for the split design, as expected. The two responses are close enough where the NOT gate response functions can be used in the UCF for circuit design without having to include data for the 2D response functions associated with NOR gates.

We found Cello's prediction of circuit response improved with the use of split gates. An example is shown in Fig 3D for a 2-input AND gate where the signals from the two sensors are inverted with NOT gates before being integrated by a NOR gate. The design based on the tandem gates shows the on state is far from the predicted response. In contrast, when the split-gate design is used, the data closely match the predicted responses (Fig 3E). In addition to the AND gate, we constructed two versions of a 3-input circuit that contains four NOT/NOR gates and compared its output to that predicted for all eight combinations of inputs (Fig 3F and Appendix Fig S9). The circuits based on split gates consistently outperformed those based on tandem gates, sometimes by orders of magnitude.

**Table 1. Response function parameters of NOT gates in the genome[a]**

| Gate name | $y_{max}$ (RPU) | $y_{min}$ (RPU) | $K$ (RPU) | $n$ |
|-----------|-----------------|-----------------|-----------|-----|
| P1_PhlF | 5.12 | 0.01 | 0.15 | 2.4 |
| Q1_QacR | 4.52 | 0.04 | 0.97 | 4.3 |
| A1_AmtR | 2.08 | 0.03 | 0.15 | 1.7 |
| B3_BM3R1 | 0.65 | 0.01 | 0.40 | 2.7 |
| F2_AmeRs | 3.69 | 0.03 | 0.13 | 1.7 |
| E1_BetI | 2.25 | 0.25 | 1.25 | 3.8 |

[a]The fit parameters are shown for a Hill equation ($y = y_{min} + ((y_{max}-y_{min})/(1 + (x/K)^n))$).

## Automated genetic circuit design for the genome

Cello is design automation software that allows a user to define the desired circuit function using the Verilog language and specify the sensors to serve as inputs. The design is mapped to a particular species, genetic location, and gate technology, all of which are contained in the user constraint file (UCF). The software designs the DNA sequence containing the circuit and predicts the response of the outputs for different combinations of input activity. The software also predicts the impact of the circuit states on the growth rates of the cell.

We constructed a new UCF (Eco2C1G3T1) to design circuits for the three landing pads in *E. coli* MG1655 that works with Cell version 2.0 (provided as Appendix File S1). The response functions for the gates, including cytometry distributions, are included along with the impact on growth ($OD_{600}$) (Fig 2D). Design constraints (gene order, orientation, and locations in the landing pads) are encoded as EUGENE rules (Oberortner *et al*, 2014; Nielsen *et al*, 2016). The split-gate designs required modifying the Cello code (Materials and Methods).

Cello was used to design five genetic circuits for the genome with up to 6 gates and 4 sensors (Fig 4). Each design was constructed as predicted with no additional DNA modifications (Materials and Methods). The circuits were constructed using the same sensor array in Landing Pad #3. All the circuits were characterized and found to closely match the predicted responses, with the exception of two high off states in 0x0B that were still fivefold lower than the lowest on state.

## Evolutionary stability and total RNAP flux

Circuits draw on different levels of host resources depending on their state. For example, each state of a logic circuit, defined by a combination of inputs, corresponds to a different combination of active promoters and expressed repressors (Canton *et al*, 2008). Therefore, each state requires a different amount of the host's RNAPs and ribosomes. It has been shown that the expression of heterologous genes decreases the growth rate of the host and provides evolutionary incentive to remove the offending DNA (Klumpp *et al*, 2009; Scott *et al*, 2010). We have observed a higher probability of circuit breakage under conditions where the total RNAP flux used by a plasmid-encoded circuit is high (Shin *et al*, 2020).

Two circuits were designed by Cello to compare the impact of carrying the circuit in the genome or a plasmid (Fig 5A). Each circuit was designed to produce the same 3-input logic operation, encoded using Verilog, but different UCFs were used to map the circuit to a DNA sequence. The first was designed using the Eco1C2G2T2 UCF for the p15a plasmid in *E. coli* DH10β and was published previously (Nielsen *et al*, 2016; Shin *et al*, 2020). The second was designed using the Eco2C1G3T1 for the *E. coli* MG1655 genome. Note that despite encoding the same logic operation, the circuits have different repressor assignments and DNA sequences. RPU is a surrogate for the RNAP flux exiting a promoter; thus, the total RNAP usage of a circuit can be estimated by summing the activities of all circuit promoters (including the sensor output promoters) (Materials and Methods). There is up to a fourfold decrease in total RNAP flux required by the

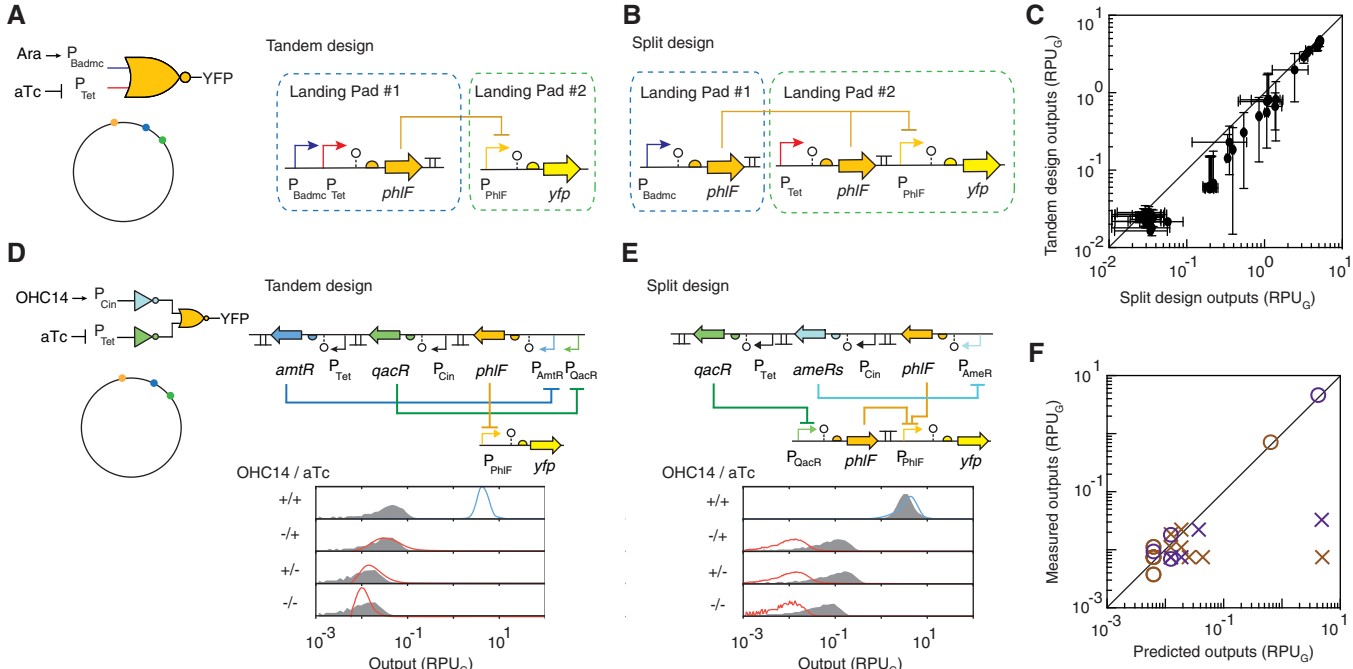

**Figure 3. Comparison of tandem and split NOT gates.**

A   Tandem NOR gate encoded on the genome. Tandem refers to the $P_{Badmc}$ and $P_{Tet}$ being encoded in series before the PhlF repressor gene, all of which are encoded within one landing pad.

B   Split NOR gate encoded on the genome. The $P_{Badmc}$ and $P_{Tet}$ promoters each drive the expression of different PhlF repressor genes and each are encoded within different landing pads.

C   A comparison of the output of the tandem and split NOR gates to changes in the input promoters. Each point represents a combination of inducers leading to the induction of the input promoters. For $P_{Bad}$ induction, 8 different concentrations of L-arabinose (0, 1 µM, 31.25 µM, 62.5 µM, 0.25 mM, 1 mM, 10 mM, and 25 mM) were used. For $P_{Tet}$ induction, 8 different concentrations of aTc (0, 0.01 0.1, 0.5, 1, 5, 10, and 20 ng/ml) were used. The line is drawn at $x = y$. The means of three experiments performed on different days are shown, and the error bars are the standard deviation of these measurements.

D   An AND gate built with a tandem NOR gate carried in Landing Pad #1. The blue and red distributions show the Cello predictions, and the gray distributions show the experimental data. The inducer concentrations are 10 µM OHC14 and 20 ng/µl aTc.

E   An AND gate built with a split NOR gate divided across Landing Pads #1 and #2.

F   The comparison between the predicted and measured outputs for different combinations of inducers for the AND gates in parts d, e (purple) and a second 3-input logic gate (Circuit 0x08, brown). Circuits based on Tandem NOR gate design are shown as symbol "X", and the circuits based on Split NOR gate design are shown as symbol "O". The additional inducer concentration is 200 mM vanillic acid (Tandem 0x08) and the genetic design, and full response of 0x08 is shown in Appendix Fig S9. The line is drawn at $x = y$. The means of three experiments performed on different days are shown, and the error bars are the standard deviation of these measurements. The error bars are often smaller than the data points.

genome-encoded circuit to function, depending on the state (Fig 5B).

The differences in resource usage lead to different impacts on cell growth between the two circuits (Fig 5C). Under our growth conditions, cells harboring genetic circuits showed lower cell growth than that of cells without the genetic circuits and that of cells only harboring a *yfp* reporter. The impact on the cell growth was lower when the circuit is carried in the genome (45% decrease in cell growth compared to cells without circuits, averaged across all eight states). When cells harbored plasmid-encoded circuits, 67% decrease in cell growth (averaged across all eight states) compared to *E. coli* DH10β with empty plasmid was observed.

The evolutionary stability of a genome-encoded circuit was then evaluated (Fig 5D). Circuits carried on plasmids can break quickly, even under antibiotic selection and when carried in recombinase-deficient strains (*E. coli* DH10β; Chen *et al*, 2013; Fernandez-Rodriguez *et al*, 2015; Shin *et al*, 2020). To replicate conditions without antibiotic selection, a p15a plasmid containing the 0xF1 circuit (Shin

*et al*, 2020) was carried for 2 weeks in *recA-E. coli* DH10β in M9 media without antibiotics. The plasmid was rapidly lost from the population, and no circuit response can be observed after 3–5 days (Fig 5A and Appendix Fig S13). In contrast, the genome-encoded circuit is stable over weeks without antibiotics while being cycled between different input states with no sign of reduced performance. After the 12-day experiment, the genome-encoded circuits were Sanger-sequenced by amplifying junctions of the circuit and genomic DNA and no mutations were observed to the circuit DNA.

## Discussion

We demonstrate that design automation can be applied to create stable regulatory networks carried in the genome of *E. coli*. This requires re-optimizing gates for the genome and re-measuring their response functions in this context. But once this is done and the data used to populate a UCF file, then designing a circuit for either a

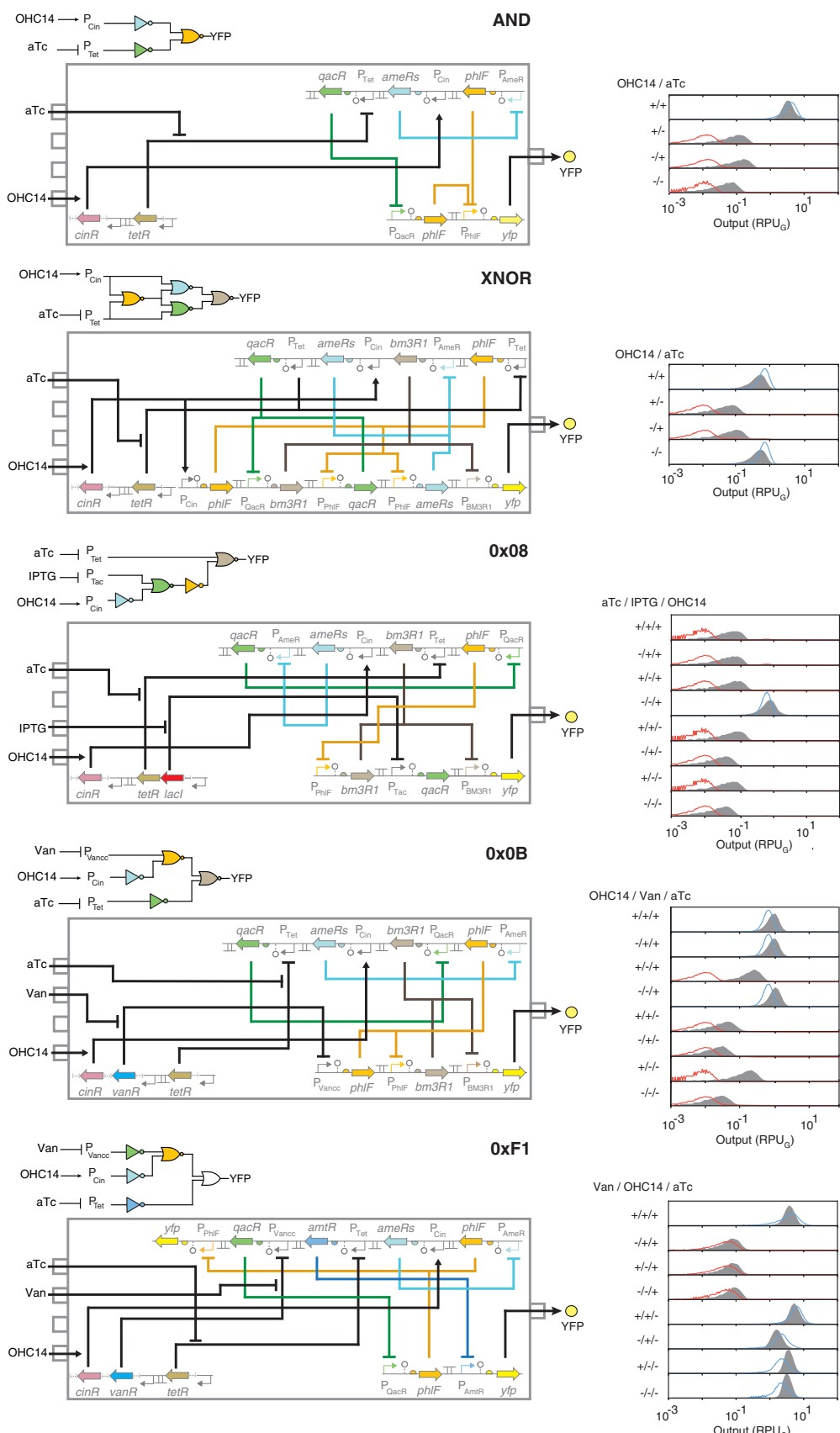

**Figure 4.**

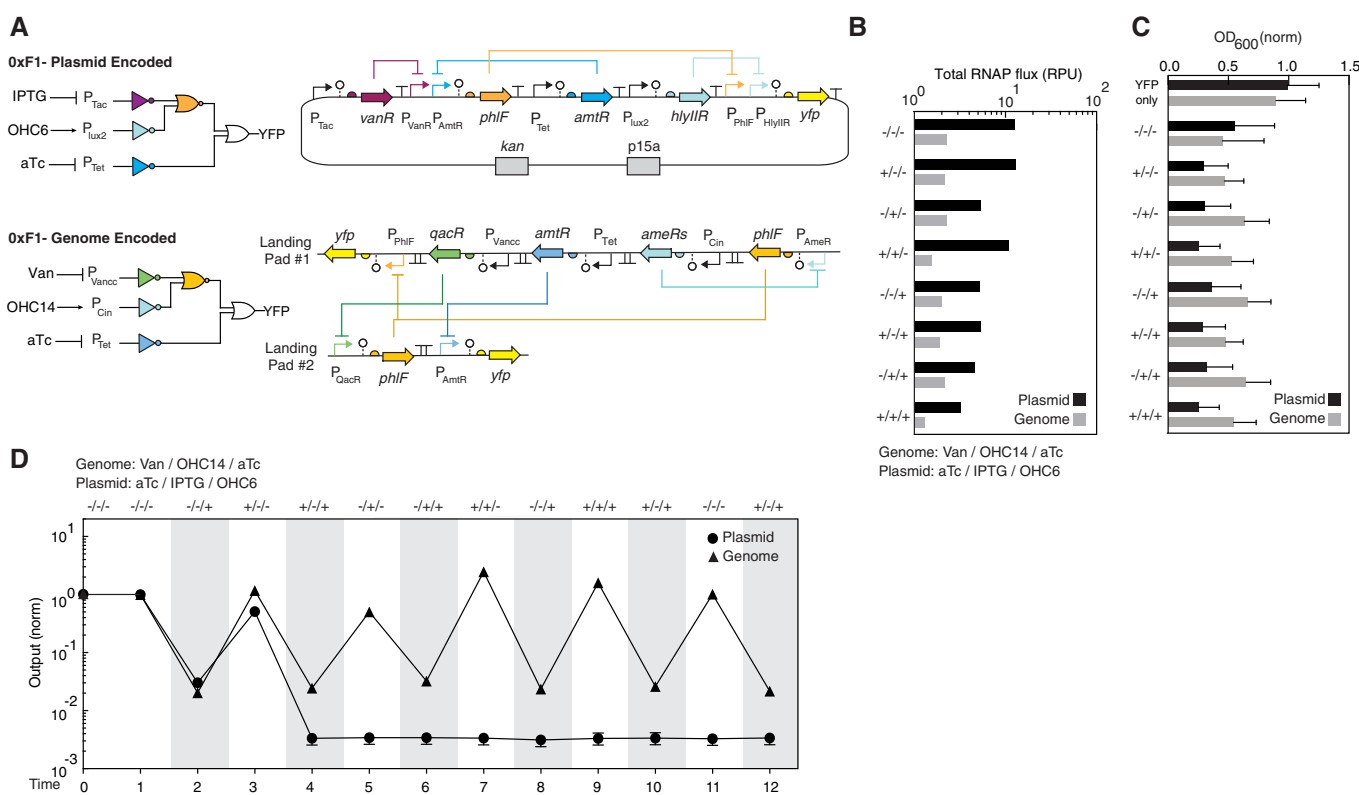

**Figure 4. Genome-encoded genetic circuits designed by Cello.**

The wiring diagrams are shown with gates colored by the repressor identity. The gray boxes show the genetic circuit, with the construct in Landing Pad #1 at the top and Landing Pad #2 at the bottom. The genetic sensors used for the circuit are shown to the bottom left; all circuits use the same sensor array (Landing Pad #3), and [. . .] indicates portions of it not visualized for clarity. The concentrations of inducers used are 1 mM IPTG, 20 ng/µl aTc, 200 mM vanillic acid, and 10 µM of OHC14. The distributions predicted by Cello are shown in blue or red, and the gray distributions are the experimental measurements. The experiments were repeated three times on different days with similar results (error bars shown in Appendix Fig S10).

**Figure 5. Evolutionary stability of a circuit encoded on a genome versus a plasmid.**

A The wiring diagrams and designs are shown. The sequences of genetic parts are provided in Appendix Table S5.

B The total RNAP flux for the genome-encoded circuit (gray) is compared to the circuit designed for a p15a plasmid (black) (Materials and Methods). The inducers for the genome-encoded circuit and plasmid-encoded circuit are 200 µM vanillic acid, 10 µM OHC14/20 ng/µl aTc and 2 ng/µl aTc, 0.2 mM IPTG and 0.1 pg/µl OHC6, respectively.

C Cell density measurements for cultures grown in the different circuit states (combinations of inducers). The means of three experiments performed on different days are shown, and the error bars are the standard deviation of these measurements. The cell densities are normalized by the $OD_{600}$ measured using cells without genetic circuits (Materials and Methods).

D The activities of the circuit output promoters are shown as a function of time. The data are normalized by the fluorescence of the first time point (time = 0). The shading indicates the periods where cells are grown in the presence of the combinations of inducers shown at the top. The inducers for the genome-encoded circuit (triangles) and plasmid-encoded circuit (circles) are 200 µM vanillic acid, 10 µM OHC14, 20 ng/µl aTc and 2 ng/µl aTc, 0.2 mM IPTG and 0.1 pg/µl OHC6, respectively. The dashed line represents the autofluorescence of wild-type *E. coli* MG1655. The circuit designs and their responses to different combinations of inducers are shown in Appendix Fig S11. The autofluorescence was measured as 0.05 (a.u.).

plasmid or the genome is as simple as selecting a different UCF. Mapping of the circuit function to DNA is completely different in these cases, resulting in different assignments of repressors to the gates. Circuits designed using this approach are found to function as predicted, but this requires (i) insulation from context effects by carrying the circuit in an insulated landing pad, and (ii) a new gate architecture that functions more reliably, albeit requiring more DNA to encode.

This allows circuits to be designed for different genetic locations, depending on the needs of the application. A responsive therapeutic or diagnostic application may require the regulation be maximally sensitive and produce the largest possible response for a limited period of time (Chowdhury *et al*, 2019). Some agricultural applications require very high expression of enzymes, for example, up to 25% of cell mass is composed of nitrogenase (Ryu *et al*, 2020). These cases benefit from plasmids stabilized with additional

systems (Easter *et al*, 1998; Prell *et al*, 2002; Fedorec *et al*, 2019). When cells have to be maintained over long periods in a competitive environment, for example, in the microbiota in the human gut or in soil in agriculture, then it is important to reduce the resource utilization(Klumpp *et al*, 2009; Liao *et al*, 2017; Riglar *et al*, 2017; Ceroni *et al*, 2018; Bloch *et al*, 2020). Similarly, in bio-production applications that are sensitive to titers and yields from feedstock, the circuit needs to have essentially no growth impact and minimal draw on carbon or energy resources. These require maintaining the minimal copy number possible, either in the genome or from a bacterial artificial chromosome (BAC).

With the ease of DNA synthesis and genome editing, entire genomes can be built (Annaluru *et al*, 2014). Increasingly ambitious efforts have been undertaken to reorganize these genomes to simplify their further engineering, for example, the removal of transposons and problematic sequences, the reorganization of genes into clusters defined by function, or even the joining of entire chromosomes to create a single sequence (Datsenko & Wanner, 2000; Baba *et al*, 2006; Hutchison *et al*, 2016; Mitchell *et al*, 2017; Wu *et al*, 2017; Xie *et al*, 2017; Shao *et al*, 2018). Here, we start to think about the industrial design behind the organization of synthetic systems into the chromosome. Placing the sensors together into one landing pad helps organize the design. Similarly, allocating landing pads for regulatory circuits has an anthropomorphic purpose. One can imagine extending this to defining dedicated spaces for metabolic pathways, stress response, or combinations of functionality required for different stages of an application. Insulating these systems from the background processes required for growth and basic cellular function, as well as predicting and measuring this impact, will be increasingly important when designing genomes for increasingly complex applications, whether it be in a fermenter, the body, or the environment.

# Materials and Methods

### Strains and media

Unless otherwise noted, the strain used for the characterization of genetic circuits is *E. coli* MG1655 (numbering is based on the genome sequence, NCBI U00096.3; Blattner *et al*, 1997). Plasmid engineering was performed using *E. coli* DH10β (New England Biolabs, USA, C3019H) or *E. coli* DH5α (New England Biolabs, USA, C2988J). *E. coli* TransforMax™ EC100D™ pir⁺ (Lucigen, USA, CP09500)) and *E. coli* JTK164A (Kittleson *et al*, 2011) were used for plasmids containing the R6K origin of replication. For conjugation, *E. coli* S17-1 λpir strain (TpR, SmR, *recA*, *thi*, *pro*, *hsdR*-M+RP4: 2-Tc:Mu: Km Tn7 λpir) was used as a donor strain to deliver plasmids with R6K origin of replication and OriT. LB media (BD Biosciences, USA, BD244610) were used for cell growth and cloning. 2xYT media (BD Biosciences, USA, DF0440-17) were used to grow cells for plasmid extraction with the QIAprep Spin Miniprep Kit (Qiagen, USA, 27104). Electrocompetent cells were prepped in SOB media (Teknova, USA, S0210). SOC recovery media (New England Biolabs, USA, B9020S) were used to recover cells after transformation. M9 media consist of M9 minimal salt (Sigma-Aldrich, USA, M6030) supplemented with 0.034% thiamine (Fisher Scientific, USA, BP892-100), 0.4% glucose (Fisher Chemical, USA, M-10046U), and 0.2%

casaminoacids (BD Biosciences, USA, 223050). This media (hereafter referred to as "M9 media") were used for all measurements and characterizations, unless noted otherwise. Antibiotics used are as follows: ampicillin (100 μg/ml, Amp) (GoldBio, USA, A-301-5), chloramphenicol (34 μg/ml, Cm) (Alfa Aesar, USA, AAB20841-14), kanamycin (50 μg/ml, Kan) (GoldBio, USA, K-120-10), spectinomycin (40 μg/mL, Sp) (MP Biomedicals LLC, USA, 158993), and tetracycline (5 μg/ml, Tet) (GoldBio, USA, T-101-25). All oligonucleotides, Gblocks, and oligos were ordered from IDT (Integrated DNA Technologies, USA; Appendix Fig S14 and Appendix Tables S4–S8).

### Terminator identification from genome sequences

Candidate terminators were identified from the *E. coli* MG1655 genome (NCBI RefSeq: NC_000913) using RNIE version 0.01 with default settings and the "genome.cm" terminator model (Gardner *et al*, 2011). This produced an output GFF file containing the location (start and end base pair), orientation (sense or antisense strand), and scoring statistics for each putative terminator part. To assess the strength of each putative terminator, transcription profiles were generated for both sense and antisense strands of the MG1655 genome from the raw RNA-seq reads (Gorochowski *et al*, 2017). Next, for each putative terminator the appropriate transcription profile for the sense or antisense strand (depending on the orientation of the terminator) was selected. The terminator strength was then estimated by measuring the average transcription profile height for the 25-bp region before and after the position of the terminator (to smooth localized fluctuations) and calculating the ratio of the average transcription profile height directly after the terminator to directly before. Finally, those terminators that had been used previously in other work (Chen *et al*, 2013) were filtered out and a final ranked list of terminators by termination strength produced.

### Measurement of terminator strength

Terminators were characterized following a previously published assay (Chen *et al*, 2013). They were cloned into the pGR plasmid (Appendix Fig S14, Appendix Fig S1, Appendix Tables S1 and S5). *Escherichia coli* DH5α harboring pGR plasmids with a terminator (pGR-DT#) were cultured overnight in 200 μl of LB medium with ampicillin (100 μg/ml). Cells were cultured using Nunc™ 96-well plates (Thermo Scientific, USA, 249662) in an ELMI Digital Thermo Microplate Shaker Incubator (ELMI Ltd, Latvia; hereafter "ELMI plate shaker"). The next day, cells were 200-fold diluted into 200 μl of fresh LB medium with ampicillin (100 μg/ml) and 12.5 mM L-arabinose (Sigma-Aldrich, USA, A3256). Cells were induced for 3 h at 37°C and 1,000 rpm in an ELMI plate shaker. After the induction, fluorescence levels were analyzed with flow cytometry and the geometric mean of GFP (FITC-A) and RFP (PE-Texas-RED-A) was calculated using FlowJo (TreeStar, Inc., USA) software. These geometric means were then used to calculate the terminator strength, $T_S = [(<GFP>_{term}/<RFP>_{term})/(<GFP>_{ref}/<RFP>_{ref})]$. The $<RFP>_{term}$ and $<GFP>_{term}$ denote geometric mean calculated for cells containing the plasmid terminator after subtracting the autofluorescence. The $<RFP>_{ref}$ and $<GFP>_{ref}$ denote geometric mean calculated with the pGR plasmid without a terminator between GFP and RFP after subtracting the autofluorescence.

## Flow cytometry analysis

Cytometry was performed using a LSRII Fortessa flow cytometer (BD Biosciences, USA). Upon harvesting, cells were diluted into 200 μl of 1× PBS ([NaCl]: 137 mM, [KCl]: 2.7 mM, [Na$_2$HPO$_4$]: 10 mM and [KH$_2$PO$_4$]: 1.8 mM) with 2 mg/ml kanamycin. An FSC voltage of 437 V, SSC voltage of 289 V, a green-laser (488 nm) voltage of 425 V, and a red-laser (561 nm) voltage of 489 V were used. For each sample, > 30,000 events were recorded. The recorded flow cytometry data were further analyzed with the software FlowJo. FITC-A and PE-Texas Red-A median values were used to represent the expression level distribution within a population.

## Construction of Tn5 transposon library

The plYJP017 plasmid that contains a constitutively expressed mCherry probe was constructed by modifying the pBAMD1-4 plasmid (Martinez-Garcia *et al*, 2014). To prevent constant *tnpA* expression from backbone integration, *sfGFP* was added to the backbone as a counter-selection marker (Fig 1b). *Escherichia coli* S17-1 λpir electrocompetent cells were transformed with the plYJP017 plasmid. The strain harboring plYJP017 plasmid was then used for conjugation with *E. coli* DH10β carrying a tetracycline (Tet) resistance marker (*tetA*) in the genome (YJP_DHC404; Appendix Fig S14). Donor strains (*E. coli* S17-1 λpir) and recipient strains (*E. coli* DH10β) were separately grown overnight in 200 μl LB media with antibiotics at 37°C and 1,000 rpm using Nunc™ 96-well plates (Thermo Scientific, USA, 249662) in an ELMI plate shaker. The next day, cells were 200-fold diluted into 4 ml LB media with antibiotics and incubated for 2.5 h at 37°C at 250 rpm in a New Brunswick Innova 44 Shaker (Eppendorf, USA). When cells reached OD$_{600}$ = 0.4, 250 μl of both donor and recipient cells were mixed. Cells were then gently centrifuged (8,000 *g*, 25°C) and washed with 1 ml of LB without antibiotics four times at room temperature. Finally, cell pellets were resuspended with 50 μl of SOC recovery media and were spotted on a plain LB agar plate (7 μl per spot). Spotted LB agar plates were incubated 5 h at 37°C. Cells were then collected and resuspended into 1 ml SOC recovery media, followed immediately by two additional dilutions (100-fold and 10,000-fold). Then, 75 μl of each diluted cell suspension was plated on LB agar plates with spectinomycin (40 μg/ml) and tetracycline (5 μg/ml). The next day, single colonies were picked from plates and grown in 200 μl M9 media for 5.5 h at 37°C and 1,000 rpm in an ELMI plate shaker using Nunc™ 96-well plates (Thermo Scientific, USA, 249662). After the incubation, cells were analyzed with flow cytometry by measuring GFP (FITC-A) and mCherry (PE-Texas RED) expression levels. Only cells that have mCherry but not GFP were used for further analysis to determine the insertion location. For each colony, the insertion location was determined by amplifying the junction between inserted DNA and the neighboring genomic DNA. A randomized primer (oYJP1741: GGCACGCGTCGACTAG-TACNNNNNNNNNNNNACGCC) and an insertion-specific primer (oYJP1745: CTTGGCCTCGCGCGCAGATCAG; Martinez-Garcia *et al*, 2014) were used to amplify the junction using two consecutive rounds of PCR amplifications. Each colony was suspended in water and was incubated at 95°C for 10 min to completely lyse cells. A 1.25 μl aliquot of the colony suspension was added as a PCR template to the PCR premix that has 12.5 μl of 2× Phusion High-

Fidelity Master Mix (New England Biolabs, USA, M0531), 10 μl of water, 1.25 μl of 10 μM oYJP1741 primer, and 0.5 μl of 10 μM oYJP1745 primers. The PCR products were then Sanger-sequenced with an internal primer oYJP1746 (CACCAAGGTAGTCGGCAAAT) and aligned using NCBI nucleotide blast. Only the insertions with a unique hit were selected for further characterization. The mCherry expression levels and growth phenotypes were characterized for each member of the Tn5 transposon library. Each clone was streaked on LB agar plates with spectinomycin (40 μg/ml) and Tet (5 μg/ml). Single colonies were picked from plates and were inoculated overnight in M9 media without antibiotics for 16 h at 37°C and 1,000 rpm in an ELMI plate shaker using Nunc™ 96-well plates (Thermo Scientific, USA, 249662). Cells were diluted 185-fold into 200 μl of fresh M9 media and incubated for 3 h. Cells were diluted again 700-fold into 200 μl of fresh M9 media and grown for 6 h. After incubating for 6 h, 30 μl of cells was added to 200 μl of 1× PBS solution with 2 mg/ml kanamycin (Kan) and fluorescence measured using flow cytometry. The optical density (OD) at 600 nm of the cultures was measured. To do this, 150 μl of the culture was transferred to an optically transparent Nunc™ 96-well plates (Thermo Scientific, USA, 165305). The Hybrid Microplate Reader BioTek Synergy H1 (BioTek Instruments Inc, USA) was used to measure the final absorbance of the culture. The relative growth for each member of the library was calculated by relative growth = $((OD_{600:\ Tn5}) - (OD_{600:\ blank}))/(OD_{600:\ DH10\beta\ Tet}) - (OD_{600:\ blank}))$, where $OD_{600:\ Tn5}$, $OD_{600:\ blank}$ and $OD_{600:\ DH10\beta\ Tet}$ refer to $OD_{600}$ of Tn5 library member, blank M9 media, and *E. coli* DH10β with the tetracycline marker on the genome, respectively.

## Computational search for putative off-target integrase sites

To identify potential off-target sites for integrase 2, 5, and 7, the published att B/P sites (Yang *et al*, 2014) (Appendix Tables S5 and S6) were searched for in the *E. coli* MG1655 genome (NCBI U00096.3) using megablast (Zhang *et al*, 2000) and blastn (Altschul *et al*, 1997). Each site was tested whether it (i) has any matches to the genome via megablast, (ii) covers > 30% of the query sequence via blastn, and (iii) has a match with significant overlap (E-value < 0.1). If any of these criteria were true, the site was rejected.

## Construction of genomic landing pads

Two methods were used to insert the landing pads into the genome. Two landing pads (#1 and #2) were introduced sequentially using λ-RED recombineering (Datsenko & Wanner, 2000). *E. coli* MG1655 cells harboring arabinose-inducible λ-RED recombinase on a plasmid with a temperature-sensitive origin (pKD46; Datsenko & Wanner, 2000) were grown overnight using Falcon 14-ml round-bottom polypropylene tubes (Corning, USA, 352059). Cells were grown in 4 ml LB with Amp (100 μg/ml) at 30°C and 250 rpm in a New Brunswick Innova 44 Shaker (Eppendorf, USA). The next day, cells were diluted 200-fold into 25 ml of fresh SOB medium in a nicked-bottom Erlenmeyer flask with ampicillin (100 μg/ml) and 10 mM L-arabinose. Cells were induced for three hours at 30°C and 250 rpm in a New Brunswick Innova 44 Shaker (Eppendorf, USA) until they reached OD$_{600}$ = 0.4. Cells were then centrifuged (4,700 *g*, 4°C, 10 min) using Legend XFR centrifuge (Thermo Scientific, USA). Cell pellets were then washed with 5 ml of chilled 10%

glycerol and spun down again at 4,700 *g* and 4°C for 10 min. The washing step was repeated two more times using 1 ml of chilled 10% glycerol, and cells were pelleted using a refrigerated benchtop centrifuge (Eppendorf, USA) at 21,000 *g* and 4°C for 30 s. Finally, cell pellets were resuspended in 500 μl of chilled 10% glycerol. Next, 200 μl of these electrocompetent cells was transformed with 150 ng of landing pad DNA using electroporation (2,500 mA) (Eppendorf, USA) and recovered by adding 1 ml SOC recovery media and incubated at 30°C for 3 h. Transformed cells were then plated on LB agar (2%) with Cm (35 μg/ml) or Kan (50 μg/ml) antibiotics and incubated at 30°C overnight. Integration of landing pads was confirmed by PCR amplification and sequencing of the genomic regions that include landing pads. For Landing Pad #1, amplifying primers oYJP3436 (CCTGATCAGGTTCCGCGGATCCC-GAATAAACGGTC) and oYJP3437 (AGGCGCTGGAAGCGCGCTTTG TGCTGGAAGATAAG) were used. For Landing Pad #2, amplifying primers oYJP3525 (ACCAATTGGCGCGCGCTTCGCAATAAAATTCC CTTCG) and oYJP3526 (TGCCAAAGGCGATAGGTGAAATAATGTC GGCGACAGCGG) were used. After integrating these two landing pads, the temperature-sensitive plasmid harboring λ-RED recombi-nase was removed by growing cells overnight at 37°C in LB without Amp (100 μg/ml). After the Landing Pads #1 and #2 were success-fully integrated, Landing Pad #3 was constructed and integrated using a site-specific mini-Tn7 transposase (Choi & Schweizer, 2006). First, the Landing Pad #3 was cloned into plasmid plYJP072 (Appendix Fig S14). Next, *E. coli* S17-1 λpir electrocompetent cells were transformed with plYJP072 and the resulting strain was conju-gated with two different strains for biparental mating. These two strains consist of a strain harboring Tn7 transposase-encoding plas-mids and the strain containing Landing Pads #1 and #2 (described above). The conjugated cells were plated on LB agar plate (2%) with Cm (35 μg/ml) or Kan (50 μg/ml) and Tet (5 μg/ml) antibiotics. The insertion of Landing Pad #3 was confirmed by PCR-amplifying and sequencing genomic regions that include the landing pad by using PCR primers oYJP2826 (AGAGATGACAGAAAAATTTT-CATTCTGTGACAGAGAAAAAGTAGCCGAAGATG) and oYJP2827 (CCGCGTAACCTGGCAAAATCGGTTACGGTTGAGTAA). After the landing pads were inserted, phage transduction was used to move them into a clean genomic background. Transduction using P1 phage followed a previously published protocol, unless otherwise noted (Thomason *et al*, 2007). To prepare P1 lysate, *E. coli* MG1655 cells harboring landing pads with three antibiotic markers (Kan, Cm, and Tet) (YJP_MKC172) were cultured overnight at 37°C in LB media with antibiotics. The next day, cells were diluted 100-fold into 5 ml LB supplemented with 0.2% glucose and 5 mM calcium chlo-ride (CaCl$_2$) without antibiotics. After 45 min of incubation at 37°C, 250 rpm in a New Brunswick Innova 44 Shaker (Eppendorf, USA), 100 μl of P1 phage stock harvested from *E. coli* MG1655 was added to the culture and incubated for 3 h at 37°C. Once the culture was cleared, a few drops of chloroform (CHCl$_3$) were added. Cell debris was spun down at 9,200 *g* for 10 min at 4°C using a refrigerated benchtop centrifuge (Eppendorf, USA). The resulting P1 lysate was further purified through 0.45-μm syringe filter (VWR international USA, 28145-481). To perform P1 transduction, wild-type *E. coli* MG1655 was grown in LB overnight at 37°C from a single colony. The next day, 1.5 ml of overnight culture was harvested and resus-pended in 0.75 ml of P1 salt solution (10 mM CaCl$_2$ and 5 mM MgSO$_4$) (Fisher Scientific, USA). Varying volumes of P1 lysate (100,

10, and 1 μl) were added to the 100 μl of the resuspended cells and were incubated for 30 min at 25°C. The mixtures were transferred into 1 ml of LB supplemented with 200 μl of sodium citrate and were incubated for 1 h at 37°C and 250 rpm in a New Brunswick Innova 44 Shaker (Eppendorf, USA). After the incubation, cells were centrifuged at 21,000 *g* and 25°C for 30 s and were plated on LB agar (2%) plates with antibiotics and 5 mM sodium citrate. Colonies were verified by PCR-amplifying genomic DNA with primers as described above (oYJP3436 and oYJP3437 for Landing Pad #1, oYJP3525 and oYJP3526 for Landing Pad #2, and oYJP2826 and oYJP2827 for Landing Pad #3). After the genomic insertion of land-ing pads, each landing pad contained a unique antibiotic resistance marker (Cm, Kan, and Tet for Landing Pads #1, #2, and #3, respec-tively). These antibiotic resistance markers were located between a pair of unidirectional flippase recognition target (FRT) sites.

## RNA-seq

RNA-seq libraries were prepared following a previously described method (Gorochowski *et al*, 2017). *Escherichia coli* MG1655 strains with integrated landing pads (YJP_MKC173) were first streaked on LB agar (2%) plates without antibiotics and incubated at 37°C. Single colonies were selected and grown overnight in M9 media without antibiotics at 37°C. The next day, cells were diluted 185-fold into 200 μl M9 media without antibiotics and grown for 3 h at 37°C and 1,000 rpm in an ELMI plate shaker using Nunc™ 96-well plates (Thermo Scientific, USA, 249662). After 3 h, cells were diluted 700-fold by adding 4.28 μl of the culture into 3 ml of M9 media without antibiotics. Cells were grown using Falcon 14-ml round-bottom polypropylene tubes (Corning, USA, 352059) in an Innova 44 Shaker (Eppendorf, USA) at 37°C, 250 rpm. After 5 h, cells were spun down at 4°C and 21,000 *g* for 3 min to collect the cell pellets for RNA-seq library preparation. After discarding the supernatants, cell pellets were flash-frozen in liquid nitrogen for storage at −80°C. Cell pellets were lysed by adding 1 mg of lysozyme, in 10 mM Tris–HCl (pH 8.0) with 0.1 mM EDTA, and total RNA was extracted using PureLink RNA Mini Kit (Life Technologies, CA, 12183020). RNA samples were further purified and concentrated with RNA Clean & Concentrator-5 Kit (Zymo Research, R1015), which was verified by Bioanalyzer (Agilent, CA). Ribosomal RNAs were depleted from RNA samples using Ribo-Zero rRNA Removal Kit for bacteria (Illu-mina, CA, MRZMB126). RNA-integrity numbers (RIN) were obtained for each sample, and only those samples with RIN > 8.5 were selected for library preparation. Strand-specific RNAtag-seq libraries were created by the Broad Technology Labs Microbial Omics Core (MOC) where uniquely barcoded samples were pooled together to run on two separate lanes of an Illumina HiSeq 2500. After the sequencing runs, reads from both lanes were combined, and the pooled mixture was de-multiplexed into original samples, followed by trimming the barcode tag from each read. Lysozyme, Tris–HCl (pH 8.0), and EDTA that were used for RNA-seq library preparation were purchased from Sigma-Aldrich (L6871), USB (75825), and USB (15694), respectively. Raw sequencing reads were aligned to the reference genomes, and transcription profiles were generated following a previously developed in-house Python script (Gorochowski *et al*, 2017). Briefly, the first step of the process was to generate new reference files for the genome of strains with inte-grated landing pads. Therefore, for each strain, all integrated DNA

sequences were inserted into their corresponding locations on the reference genome of *E. coli* MG1655 (NCBI RefSeq: NC_000913.3). These new FASTA and GFF files were then used to perform the alignment of raw reads using BWA version 0.7.4 with default settings, resulting in corresponding SAM and BAM files. Next, BAM files were filtered using the "view" command of SAMtools (Li *et al*, 2009; Barnett *et al*, 2011), and filter codes 83 and 163 were applied to select the reads mapping to the sense strand, and filter codes 99 and 147 were applied to select reads mapping to the antisense strands. Finally, filtered read coverage at each position along the reference sequence was normalized by the total mapped nucleotides across the genome and multiplied by $10^9$ to generate the transcription profiles in both forward and reverse directions across the genome.

### Evaluation of thiamine-dependent growth

*Escherichia coli* MG1655 and two strains harboring Landing Pad #1 v1 and v2 were grown in M9 media (with thiamine) overnight. The next day, all three cultures were centrifuged (15,000 *g*, 25°C, 3 min) and resuspended into DI water three times to remove residual thiamine in the media. The $OD_{600}$ of resuspended cells was measured, and the cells were diluted to $OD_{600} = 0.01$. Each dilution was then inoculated into thiamine-free M9 medium, consisting of M9 minimal salt (Sigma-Aldrich, USA, M6030) supplemented with 0.4% glucose and 0.2% casaminoacids (BD Biosciences, USA, 223050). After 6 h of incubation at 37°C and 250 rpm in a New Brunswick Innova 44 Shaker (Eppendorf, USA), the $OD_{600}$ of three samples and a blank sample containing only thiamine-free M9 media were measured using a Cary 50 Bio Spectrophotometer (Agilent, USA).

### Insertion of payloads into landing pads

Note that an easy-to-follow detailed protocol is provided as Appendix Note S1. The strain containing the empty landing pads was co-transformed with a plasmid encoding three integrases (plYJP053) and a plasmid containing the DNA payloads (plYJP066-KanR, plYJP070-CmR, and plYJP064-TetR). To prepare electrocompetent cells, a single colony was inoculated into 2 ml LB without antibiotics and grown for 12 h at 37°C and 250 rpm in a New Brunswick Innova 44 Shaker (Eppendorf, USA) using Falcon 14-ml round-bottom polypropylene tubes (Corning, USA, 352059). The next day, 125 μl of the overnight culture was added to 25 ml SOB medium in a nicked-bottom Erlenmeyer flask. Cells were grown for 2 h at 37°C and 250 rpm in a New Brunswick Innova 44 Shaker (Eppendorf, USA). When the early exponential phase was reached ($OD_{600} = 0.3$–0.5), cells were centrifuged (4,700 *g*, 4°C, 10 min) and washed with chilled 10% glycerol three times. After the third wash, cells were resuspended with 200 μl of chilled 10% glycerol. Cells were electroporated with 500 ng of plYJP053 plasmid and 500 ng of the payload plasmids. Immediately after the transformation, 1 ml of SOC recovery media was added to the cells. Cells were then incubated at 30°C for 3 h and plated on LB agar plates (2%) with necessary antibiotics. The insertion into the Landing Pads #1, #2, and #3 was selected with Kan (50 μg/ml), Cm (35 μg/ml), and Tet (5 μg/ml), respectively. To confirm the integration with colony PCR, primers that can amplify the junction between integrated constructs and the adjacent genomic DNA were used. For Landing Pads #1 and #2, oYJP2164 (AATAAACAAATAGGCATGGTCTAAGAAACCATT) was

used as a primer that binds to the integrated construct. oYJP3436 (CCTGATCAGGTTCCGCGGATCCCGAATAAACGGTC) and oYJP3526 (TGCCAAAGGCGATAGGTGAAATAATGTCGGCGACAGCGG) were used as primers that bind genomic DNA adjacent to Landing Pad #1 and Landing Pad #2, respectively. For Landing Pad #3, a forward primer that binds to the end of integrated construct in a forward direction was used with primer oYJP2826 (AGAGATGACA-GAAAAATTTTCATTCTGTGACAGAGAAAAAGTAGCCGAAGATG) to amplify the junction between Landing Pad #3 and the adjacent genomic DNA. The amplicon size was confirmed using gel electrophoresis. All three markers can be removed by transforming strains harboring the landing pads with inserted payloads with the pE-FLP plasmid containing a temperature-sensitive origin of replication (St-Pierre *et al*, 2013). To do this, each strain was cultured overnight in 2 ml of LB media with corresponding antibiotics (Cm (35 μg/ml), Kan (50 μg/ml), and Tet (5 μg/ml)) at 37°C and 250 rpm in a New Brunswick Innova 44 Shaker (Eppendorf, USA) using Falcon 14-ml round-bottom polypropylene tubes (Corning, USA, 352059). The next day, cells were 200-fold diluted into 4 ml of LB media without antibiotics and incubated at 37°C and 250 rpm in a New Brunswick Innova 44 Shaker using Falcon 14-ml round-bottom polypropylene tubes (Corning, USA, 352059) for 2 h until reaching $OD_{600} = 0.4$. Cells were then harvested with centrifugation (4,700 *g*, 4°C, 10 min) using a Legend XFR centrifuge. Cell pellets were washed with 1 ml chilled 10% glycerol and spun down using a refrigerated benchtop centrifuge (Eppendorf, USA) at 21,000 *g* and 4°C for 30 s for three times. Finally, cell pellets were resuspended into 75 μl of chilled 10% glycerol, yielding electrocompetent cells that were then transformed with 20 ng of pE-FLP (St-Pierre *et al*, 2013), followed by recovery in 1 ml SOC media and incubation at 30°C for 30 min. Cells were then plated on LB agar plates (2%) with Amp (100 μg/ml) and incubated at 30°C overnight. The next day, three individual colonies were streaked on LB agar plates (2%) with no antibiotics and incubated at 37°C overnight. Three colonies from each streak were then grown in LB media with four antibiotics (Amp (100 μg/ml), Cm (35 μg/ml), Kan (50 μg/ml), and Tet (5 μg/ml)) at 37°C and 1,000 rpm for 16 h on an ELMI plate shaker using Nunc™ 96-well plates (Thermo Scientific, USA, 249662). Cells that did not grow in all four antibiotics were streaked and then used as the final strains.

### Calculation of RPU (relative promoter units)

*Escherichia coli* MG1655 containing the $RPU_G$ reference promoter (YJP_MKC254) was streaked on a LB agar plate without antibiotics and incubated overnight at 37°C. Single colonies picked from the plate were then inoculated into 200 μl of M9 media without antibiotics and were incubated overnight. All culturing steps were carried out at 37°C and 1,000 rpm in an ELMI plate shaker using Nunc™ 96-well plates (Thermo Scientific, USA, 249662). The next day, cells were diluted 185-fold into 200 μl of M9 media without antibiotics and incubated for 3 h. Cells were then diluted again 700-fold into 200 μl of M9 media without antibiotics and were incubated for 5 h. Then, a 30 μl aliquot was transferred to 200 μl of 1× PBS solution with 2 mg/ml kanamycin and evaluated using flow cytometry. To convert the fluorescence of a promoter from au ($<YFP>_{measured}$) to $RPU_G$, the following equation is used: $[(<YFP>_{measured})-(<YFP>_{blank})]/[(<YFP>_{RPU})-(<YFP>_{blank})]$, where $<YFP>_{blank}$ is autofluorescence of wild-type *E. coli* MG1655.

## RPU$_G$-to-RNAP flux conversion

RPU$_G$ was converted into RNAP flux by multiplying a previously calculated conversion factor 1 RPU = 0.019 RNAP/s per DNA (B. Shao, J. Rammohan, D.A. Anderson, N. Alperovich, D. Ross & C.A. Voigt, unpublished data) and the copy number of DNA where the Landing Pad #1 is located. The copy number of the Landing Pad #1 was estimated to 3.5 by comparing the Tn5 expression data for site 7 where Landing Pad #1 is located and the site #3, a site adjacent to the single-copy region of the genome. Note that the same promoter is used for the RPU and RPU$_G$ standard cassette, and it is assumed that this promoter produces the same constitutive flux in both locations. Therefore, 1 RPU$_G$ was converted into the 0.067 RNAP/s.

## Sensor characterization

The strain containing the seven sensors in Landing Pad #3 (YJP_MKC174) was transformed with a reporter plasmid containing a promoter fused to *yfp* (plYJP067-(promoter name)) that is responsive to the seven regulators (AraC, LacI, TetR, CymR$^{AM}$, VanR$^{AM}$, CinR$^{AM}$, and TtgR$^{AM}$) and streaked on LB agar (2%) plates with Kan (50 µg/ml). Single colonies were inoculated into 200 µl M9 media with Kan (50 µg/ml) were grown overnight at 37°C and 1,000 rpm in an ELMI plate shaker using Nunc™ 96-well plates. The next day, cells were diluted 185-fold into 200 µl fresh M9 media without any antibiotics and were incubated for three hours at 37°C and 1,000 rpm in an ELMI plate shaker using Nunc™ 96-well plates (Thermo Scientific, USA, 249662). Cells were then diluted 700-fold into 200 µl fresh M9 media (no antibiotics) with appropriate inducers and were incubated for 5.5 hours in an ELMI plate shaker at 37°C and 1,000 rpm using Nunc™ 96-well plates (Thermo Scientific, USA, 249662). Then, either 12.5 mM L-arabinose (Sigma-Aldrich, USA, A3256), 1 mM IPTG (GoldBio, USA, I2481C), 20 ng/µl aTc (Sigma-Aldrich, USA, 37919), 500 µM 4-isopropylbenzoic acid (Sigma-Aldrich, USA, 268402), 200 µM vanillic acid (Sigma-Aldrich, USA, H36001), 10 µM OHC14 (Sigma-Aldrich, USA, 51481), or 1 mM naringenin (Sigma-Aldrich, USA, N5893) was used to induce the sensors. After 5.5 h, 30 µl of cells was added to 200 µl 1× PBS with 2 mg/ml Kan for flow cytometry analysis.

## NOT/NOR gate characterization

Each strain containing a NOT gate was streaked on the LB agar (2%) plates with Kan (50 µg/ml) and Cm (35 µg/ml) antibiotics. A single colony was picked and inoculated into M9 media with Kan (50 µg/ml) and Cm (35 µg/ml) for overnight culture in an ELMI plate shaker at 37°C and 1,000 rpm. Cell cultures were performed using Nunc™ 96-well plates (Thermo Scientific, USA, 249662). The next day, cells were 185-fold diluted into 200 µl fresh M9 media with no antibiotics and incubated for 3 h in an ELMI plate shaker at 37°C and 1,000 rpm. After the 3 h, cells were then diluted again 700-fold into 200 µl fresh M9 media with no antibiotics and were incubated with inducers for additional 5.5 h in an ELMI plate shaker at 37°C and 1,000 rpm. After the 5.5 h, 30 µl of cells was added to 200 µl 1× PBS with 2 mg/ml Kan for flow cytometry analysis. To measure OD, 150 µl aliquots were transferred to an optically transparent Nunc™ 96-well plates (Thermo Scientific, USA, 165305) to measure OD$_{600}$ using a Hybrid Microplate Reader BioTek Synergy H1 (BioTek Instruments Inc, USA). To measure the gate response functions, input and output promoter activities measured as median YFP fluorescence were converted into RPU$_G$. The response functions were fit to a Hill equation $y = y_{min} + ((y_{max} - y_{min})/(1 + (x/K)^n))$, using an in-house Python script.

## Circuit design automation using Cello 2.0

The Cello 2.0 software (cellocad.org) and code are available open source on GitHub (github.com/CIDARLAB/Cello-v2). Note that the UCF in this paper is designed for Cello 2.0 and will not run with the old Web-based Cello interface. A UCF file (Eco2C1G3T1.UCF.json) (Appendix File S1) was created to encode Hill parameters for NOT gate response functions in the gate library (Table 1), cytometry distributions of each gate, Eugene rules, landing pad location information, and growth assay conditions. Sensor output promoter activities (P$_{Badmc}$, Y$_{min}$ = 0.04, Y$_{max}$ = 3.33; P$_{Tac}$, Y$_{min}$ = 0.02, Y$_{max}$ = 4.20; P$_{Tet}$, Y$_{min}$ = 0.02, Y$_{max}$ = 5.41; P$_{CymRC}$, Y$_{min}$ = 0.19, Y$_{max}$ = 2.39; P$_{VanCC}$, Y$_{min}$ = 0.02, Y$_{max}$ = 3.79; P$_{Cin}$, Y$_{min}$ = 0.01, Y$_{max}$ = 4.38 and P$_{TtgR}$, Y$_{min}$ = 0.01, Y$_{max}$ = 0.22 (in RPU$_G$)), a UCF file, a truth table formulated as a Verilog file, and the growth score cutoff (set to 0.75) were used for each circuit design. Cello 2.0 was run locally using the Cello 2.0 API.

## Genetic circuit construction

Genetic circuits were first split and cloned into two plasmids, plYJP066 (KanR) and plYJP070 (CmR), that target Landing Pads #1 and #2, respectively, using Type II assembly as previously described (Nielsen *et al*, 2016; Shin *et al*, 2020). The order of transcription units within a circuit was assigned by Cello 2.0 based on EUGENE rules. In brief, each transcription unit of the circuit was sub-cloned into plYJP080 (AmpR) (Appendix Fig S14 and Appendix Table S6) backbones with p15a origin using Type II assembly with BsaI (New England Biolabs, USA, R3733) and T4 ligase HC (Promega, USA, M1794). The reaction mix was prepared by adding 40 fmol of each DNA fragment, 1 µl of BsaI, 0.5 µl of T4 ligase HC, 1.5 µl of T4 ligase buffer (Promega, USA, M1794), and water up to 15 µl. The reaction mix was then cycled between 37°C (5 min) and 16°C (3 min) for 30 times, resulting in transcription units (in plYJP080 plasmids) that are ready to be used in genetic circuit construction. For every designed genetic circuit, all the used transcription units were assembled into the full circuit using BbsI (New England Biolabs, USA, R3539) and T4 ligase HC (Promega, USA, M1794). Reaction mixture included 40 fmol of each plYJP080 plasmids containing the transcription unit, 1 µl of BbsI, 0.5 µl of T4 ligase HC, 1.5 µl of T4 ligase buffer (Promega, USA, M1794), and water up to 15 µl. The reaction mix was then cycled between 37°C (5 min) and 16°C (3 min) for 50 times for assembly reaction, resulting in two new plasmid backbones plYJP066 and plYJP070 plasmids. These two plasmids were then integrated into the genome as described above. Antibiotic markers were removed by transforming pE-FLP (St-Pierre *et al*, 2013). pE-FLP plasmid was cured by incubating cells at 37°C as previously described (Appendix Fig S14).

## Genetic circuit characterization

Strains harboring genetic circuit were streaked on the LB agar (2%) plate without antibiotics. *E. coli* MG1655 wild-type and $RPU_G$ standard strains were streaked as controls. Individual colonies were picked from plates and were inoculated into M9 media without antibiotics. Cells were incubated overnight at 37°C and 1,000 rpm in an ELMI plate shaker using Nunc™ 96-well plates (Thermo Scientific, USA, 249662). The next day, cells were then diluted 185-fold into 200 μl of fresh M9 media without antibiotics and incubated for three hours at 37°C, 1,000 rpm in an ELMI plate shaker using Nunc™ 96-well plates (Thermo Scientific, USA, 249662). After three hours, cells were 700-fold diluted into M9 media without antibiotics and were induced with appropriate combinations of inducers such as 1 mM IPTG, 12.5 mM L-arabinose, 20 ng/μl aTc, 10 μM OHC14, and 200 μM vanillic acid were used as indicated in Cello 2.0 prediction. After 5.5 h, 30 μl of cells was added to 200 μl 1× PBS with 2 mg/ml Kan for flow cytometry analysis. Median YFP fluorescence from each sample was analyzed using FlowJo (TreeStar, Inc., USA) software and was converted into $RPU_G$.

## Calculation of total RNAP flux

The input promoter activity (RNAP flux) of every NOT and NOR gate in the 0xF1 circuit was calculated for each state using the Cello 2.0 software package. The total RNAP flux for a circuit was calculated by summing the promoter activities across all the gates in the circuit, including the output promoters of the sensors. For the genome-encoded 0xF1 circuit, we used the UCF file Eco2C1G3T1, the truth table (0xF1.v), and the genome-encoded sensor output promoter activities (described above). The UCF was based on an ordinary additive model for tandem promoter activity and was not encoded with tandem roadblocking rules. For the plasmid-encoded 0xF1 circuit, we used the UCF file Eco1C2G2T2, the truth table (0xF1.v), and the plasmid-encoded sensor output promoter activities ($P_{lux2}$, $Y_{min} = 0.030$, $Y_{max} = 2.234$; $P_{Tac}$, $Y_{min} = 0.018$, $Y_{max} = 1.689$; $P_{Tet}$, $Y_{min} = 0.040$, $Y_{max} = 1.967$; and $P_{Cin}$, $Y_{min} = 0.005$, $Y_{max} = 3.178$ (in RPU)). The UCF was encoded with a non-additive tandem promoter model. Once the circuits were designed, the design output files (0xF1_A000_logic_circuit.txt (genome), 0xF1_A001_logic_circuit.txt (plasmid)) were used to calculate total RNAP flux used by the circuit. From each file, the input promoter activity of every NOT and NOR gate in the circuit (second to the last column) was summed to calculate the total RNAP flux. The total RNAP flux calculated for genome-encoded circuits (in $RPU_G$) was then divided by 6.33 (Appendix Fig S4) to convert the $RPU_G$ into RPU. The total RNAP flux for a plasmid-encoded circuit was calculated in RPU.

## Long term stability test

The plasmid-encoded 0xF1 circuit was designed using the UCF for the p15a plasmid (Eco1C2G2T2) and was constructed in *E. coli* DH10β (Nielsen *et al*, 2016; Shin *et al*, 2020). The *E. coli* MG1655 strain harboring the genome-encoded 0xF1 circuit was streaked on LB agar (2%) plates with Kan (50 μg/ml) and Cm (35 μg/ml) antibiotics and grown overnight. The *E. coli* DH10β strain harboring the plasmid-encoded 0xF1 circuit was streaked on LB agar (2%) plates containing Kan (50 μg/ml) and grown overnight. For

each, three colonies were picked and grown in M9 media without antibiotics. Every day, each culture was diluted $10^4$-fold into 500 μl of fresh M9 media in 96-deep well plates (USA Scientific, USA, 1,896–2,000) with the inducer combination indicated in Fig 5. After incubation for 8 h at 37°C and 900 rpm in a Multitron Pro Incubator Shaker (In Vitro Technologies, VIC, Australia), 30 μl of cells was added to 200 μl 1× PBS with 2 mg/ml Kan for flow cytometry analysis and 100 μl of cells was mixed with 80% autoclaved glycerol (VWR chemical BDH1172-1LP) and stored at −80°C. Another aliquot was diluted 100-fold into fresh media with the same inducer combinations and incubated overnight. The cycle continued for 12 days by repeating this protocol. To measure the $OD_{600}$ of genome and plasmid-encoded circuits, individual colonies from the streak were inoculated into M9 media and incubated overnight at 37°C and 1,000 rpm in an ELMI plate shaker. Cells harboring plasmid-encoded genetic circuits were incubated overnight with Kan (50 μg/ml). Genome-encoded circuits were incubated overnight without antibiotics. The next day, each culture was diluted 185-fold into 3 ml of fresh M9 media and was grown for 3 h at 37°C and 250 rpm in an Innova Shaker. Three hours later, the $OD_{600}$ of each culture ($OD_{600}$) was measured using a Cary 50 Bio Spectrophotometer (Agilent, USA). The measured $OD_{600}$ was used to calculate the amount of the sample required to transfer the same number of cells to the second dilution. Cells were diluted ~700-fold into 3 ml M9 media with appropriate inducers and were grown for 5.5 h at 37°C and 250 rpm in an Innova Shaker. The growth of cells without a circuit was determined using either *E. coli* MG1655 with empty landing pads (YJP_MKC173) or *E. coli* DH10β harboring a p15a plasmid with a Kan (50 μg/ml) resistance gene (pYJP018).

# Data availability

The datasets and computer code produced in this study are available in the following databases: UCF information (Eco2C1G3T.UCF, Eco2C1G3T.input, and Eco2C1G3T.output) is available in Dataset EV1. Codes used to process the data are available on GitHub (www.github.com/CIDARLAB/Cello-v2).

**Expanded View** for this article is available online.

### Acknowledgements
This work was supported by US Defense Advanced Research Projects Agency (DARPA) 1KM and SD2 awards HR0011-15-C-0084 and FA8750-17-C-0229 (C.A.V., A.E.B., and Y.P.), National Science Foundation Award CCF-1807575 (C.A.V.), U Colorado-Boulder/Dept of Energy subaward DE-SC0018368 (C.A.V., J.S.), and a Samsung Scholarship (Y.P.).

### Author contributions
YP and CAV conceived the study and designed the experiments; YP performed all the experiments; JS designed and constructed plasmid-based genetic circuits used in the study; YP and TEG developed new terminator prediction pipeline; YP and AEB performed the RNA sequencing data analysis; and YP, AEB, and CAV wrote the manuscript with input from all the authors.

### Conflict of interest
The authors declare that they have no conflict of interest.

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
