## [Review Process File · Molecular Systems Biology]

Precision design of stable genetic circuits carried in highly-insulated *E. coli* genomic landing pads

Yongjin Park, Amin Borujeni, Thomas Goroehowski, Jonghyeon Shin, and Christopher Voigt
DOI: [10.15252/msb.20209584](https://doi.org/10.15252/msb.20209584)

Corresponding author(s): Christopher Voigt (cavoigt@gmail.com)

Review Timeline:	Submission Date:	18th Mar 20
	Editorial Decision:	20th Apr 20
	Revision Received:	2nd Jul 20
	Editorial Decision:	3rd Jul 20
	Revision Received:	7th Jul 20
	Accepted:	22nd Jul 20

Editor: Maria Polychronidou

Transaction Report:

Thank you again for submitting your work to Molecular Systems Biology. We have now heard back from the three referees who agreed to evaluate your study. Overall, the reviewers are quite supportive. They raise however some concerns, which we would ask you to address in a revision. The reviewers' recommendations are rather clear and I think there is no need to repeat any of the points listed below. Please let me know if you would like to discuss any of the issues raised.

On a more editorial level, we would ask you to address the following:

REFEREE REPORTS

Reviewer #1:

Park et al. present a high-quality synthetic biology manuscript showing an innovative and substantial improvement on the Voigt Lab's Cello gene circuit design system. Anyone familiar with synthetic biology advances will have knowledge of the Cello CAD system, and this manuscript now describes a substantial body of work that allows automated design of gene circuit that are genomically encoded in E.coli (rather than on plasmids). This offers better stability, long-term performance without antibiotic selection, and the ability to construct larger circuits with more orthogonal external inducers.

I am familiar with Cello CAD and the impressive synthetic biology design and experimental advances that were required to achieve the automation of circuit design. I find that this manuscript is similarly impressive in those regards as well as being well written and expertly presented. I struggled to find any criticisms except for typos that will probably be caught in copy-editing stages anyway. So right now I would recommend this publication with no scientific changes required. It is a great paper and one that your audience of synthetic biology readers would appreciate.

Typos:

The appendix sometimes says "Appendix Fig X" and sometimes says "Appendix SFig X"

There is inconsistency through the paper and figure legends on whether or not there is a space between a number and its units

The references list has many citation journal names beginning with a '%'

The section 'Sensor array in Landing Pad #3' ends with a reference to Fig 1c but I think this should be Fig 2c.

Reviewer #2:

Summary and general comments

In this manuscript, the authors aimed to design stable genetic circuits in the genome automatically. They constructed three genetic landing pads flanked by double terminators at the E. coli genome with high expression levels and no impact on growth. Circuit design was modularized by putting sensors and genetic circuits in different landing pads. NOT/NOR gate design based on repressors in the genome were optimized and characterized. The Cello software was used for circuit design and prediction. A new user constraint file was constructed for the new genetic landing pads. The genome encoded circuits' cost is lower than the circuits in plasmid, and circuits carried on the genome keep stable for weeks, whereas those on the plasmid break quickly.

Constructing an evolutionary-stable genetic circuit is not a trivial effort, even though many studies have reported genome-encoded genetic circuits. Also, attempt to standardize the design and introduction of genetic circuits onto the genome are worthwhile. This manuscript is important and rigorous. It makes at least two important contributions: 1) it generated landing pads to accommodate the genetic circuits in the genome, and 2) it extended the circuit design automation from the plasmid to the genome.

Major points

1. In this manuscript, the criterion to select landing pad is 1) high expression and 2) no growth defect. In this case, why didn't the authors choose site 13 (position, 3942414), which showed the highest expression level and no growth impact? More importantly, the authors hypothesized that higher expression is better. However, there's no evidence throughout the manuscript to support this notion. Also, the differences between expression levels at different genomic loci are not huge. The manuscript *only* reported 3 landing pads not only limited its application but also lower the level of its generality. Namely, people can put landing pad wherever in the genome; the location has little effects on the circuit performance.?
2. I'm not convinced of the method (OD₆₀₀) to measure growth, and the corresponding conclusions of the growth impacts. Dilution once from the overnight seed culture causes a lot of problems on the bacterial physiology. The OD₆₀₀ cannot represent the growth in this way, not talking about the "growth rate", which was used in the software.
3. In Figure 5, the evolutionary stability of the genetic circuit was investigated in a changing environment (different signal combination) and it only lasted one-day for each condition. I am very curious about the stability of the circuit in a constant condition for longer time? more days.
4. In the abstract, the authors stated that "these circuits require 8-fold less RNA polymerase when carried on a plasmid". I didn't see the experimental results.

Minor points

1. Procedure for multiple payload insertion into the genome was claimed as "rapid genome engineering methodology", but it is time-consuming to insert the payload one by one. There may/should be more efficient way(s) to do it.
2. "there was a severe growth defect when grown in media lacking thiamine (Figure 1d)."
> "Figure 1d" should be "Figure 1c"
3. "That was found to recover the transcription of the thi operon and overcome the growth defect (v2, Figure 1d)"
>"v2, Figure 1d" should be "v2, Figure 1c".
4. "These sensors produce a 12- to 640-fold induction (Figure 2c, Appendix Figures 8 and 9) with low off states and no evidence of crosstalk (Figure 1c and Appendix Figure 7)."
>"Figure 1c" should be "Figure 2c"
5. Appendix Figure 10b, the first 2 panels on the left column are the same.

Reviewer #3:

In this study, Voigt and colleagues demonstrated a systematic approach to identify and insulate "landing pads" in the E. coli genome for building complex gene circuits. The major considerations include optimal gene expression (using a transposon-guided reporter gene) and minimal growth impact and insulation by using strong transcriptional terminators. They demonstrate construction of a number of gene circuits that perform as designed using the optimal landing pads.

The study builds on the foundation of a number of conceptual and technical advances made in the past several years, some of which having been championed by the Voigt lab. These include large-scale component standardization and quantification, insulation, RNAP flux for component quantification and abstraction, computer aided design and assembly of complex gates. The major distinguishing feature of the current study is the integration of these components (and their quantitative information) for the construction of large-scale circuits, directly in the genome. The work is thoroughly carried out and sets high technical standards for circuit engineering and validation. I have only a few comments that I believe the authors address in a minor revision.

While I appreciate the technical advance demonstrated in the study, I have reservations about some of the sweeping criticisms the authors made about circuit engineering on plasmids. I am actually skeptical about some of them -- for instance, the growth impact of a circuit in a plasmid is not necessarily greater than that in the genome. It's just a matter of how the circuit is specifically engineered (e.g. plasmid copy number, circuit complexity, and choice of circuit components). Despite the best effort to insulate, an argument can be made that insertion of a large segment of DNA in the chromosome would always create an interruption (to various degrees) to the normal operation of the cell.

My point is not to diminish what the authors have achieved. But I believe they should tone down the argument for advantages of integrating circuits into the chromosomes. What's better really depends on specific engineering objectives, which vary case by case. As noted by the authors, the major objective was "to simplify the process of genome engineering so that it approaches the ease of plasmid manipulations ...". This is an important and valuable objective that the authors have amply demonstrated.

Other points:

1. I do not think the comparison between a circuit in a plasmid carrying a p15 origin (without

selection) and one in the chromosome is fair. As noted by the authors, in the absence of antibiotic selection, many natural plasmids rely on other mechanisms to ensure stable maintenance. One prime example is the F plasmid, a ~100kb plasmid that can be stably maintained in standard lab conditions without positive selection.

Again, I don't think the authors need to make this point to make the work valuable. If the authors do want to make a more general claim, as the writing appears to imply, a more rigorous experimental design is needed (where a circuit is optimized both on a plasmid and in the chromosome).

2. When reporting the growth impact of different landing pads, the authors reported the OD values. It is not clear if what stage of the cultures these values correspond to. These values are useful to report; however, if the authors intend to make strong arguments based on the growth effects (or the lack of), it is more informative to show the growth rates of the cells carrying various landing pads. Two cultures with drastically different growth rates can reach the same OD when fully grown.

3. A major design strategy is the use of strong terminators to insulate. Because of this, it is useful to explain the quantification of these terminators in a greater depth in the results section. For example, Ts was defined in the methods section but it would be helpful to define it in the main text and to elaborate a bit more how it should be interpreted.

Reviewer #1:

1. *Typos: The appendix sometimes says "Appendix Fig X" and sometimes says "Appendix SFig X" There is inconsistency through the paper and figure legends on whether or not there is a space between a number and its units. The references list has many citation journal names beginning with a '%' The section 'Sensor array in Landing Pad #3' ends with a reference to Fig 1c but I think this should be Fig 2c.*

The typos have been fixed.

Reviewer #2:

1. *In this manuscript, the criterion to select landing pad is 1) high expression and 2) no growth defect. In this case, why didn't the authors choose site 13 (position, 3942414), which showed the highest expression level and no growth impact? More importantly, the authors hypothesized that higher expression is better. However, there's no evidence throughout the manuscript to support this notion. Also, the differences between expression levels at different genomic loci are not huge. The manuscript *only* reported 3 landing pads not only limited its application but also lower the level of its generality. Namely, people can put landing pad wherever in the genome; the location has little effects on the circuit performance?*

There is a 3rd criterion. The biggest problem with landing pad site selection is the disruption of native gene expression. Although site 13 yields high expression, it disrupts *uhpT* which has a role in glucose-6-phosphate. When this occurred, we would attempt to move the landing pad within a 10kb window. However for site 13, we could not identify a position that did not disrupt endogenous gene expression.

The gene expression levels at different loci (in our hands, 3-fold) are sufficient to alter the predictions of gene circuit design. We do include experiments to this end. Appendix Figure 11 shows how changing the PhIF RBS impacts the response function. Note that the difference in RBS strengths is about 3-fold and this leads to quantitatively different response functions (Appendix Figure 7). A 3-fold change may not make a difference in its function as a "NOT gate," but it changes the response function, which makes an enormous difference in the ability for design automation to connect gates and predict their quantitative response.

2. *I'm not convinced of the method (OD₆₀₀) to measure growth, and the corresponding conclusions of the growth impacts. Dilution once from the overnight seed culture causes a lot of problems on the bacterial physiology. The OD₆₀₀ cannot represent the growth in this way, not talking about the "growth rate", which was used in the software.*

We agree and have clarified the Methods. OD₆₀₀ measurements are made with a double-back-dilution protocol. The exception to this is the experiment shown in Figure 1c, we had to use a single back dilution from overnight culture. This is due to the *thiC* KO strain's slow growth in thiamine-deprived media. The *thiC* KO strains grew so slowly that a second dilution was not practical. Note that this observation supports the intent of showing these data to highlight the impact of this mutant on growth.

3. *In Figure 5, the evolutionary stability of the genetic circuit was investigated in a changing environment (different signal combination) and it only lasted one-day for each condition. I am very curious about the stability of the circuit in a constant condition for longer time/more days.*

We tested the evolutionary stability of genetic circuits shown in Figure 5 maintaining constant inducer concentrations (thus keeping the circuit state) up to 14 days. We have added an Appendix Figure 13 to demonstrate the stability of genome-encoded circuits over the time during this growth experiment.

4. *In the abstract, the authors stated that "these circuits require 8-fold less RNA polymerase when carried on a plasmid". I didn't see the experimental results.*

We have added Figure 5b to show the reduction in RNAP flux that occurs when the circuit is encoded in the genome. Note that we reduced the number to 4-fold to match the specific circuit in this figure.

5. *Procedure for multiple payload insertion into the genome was claimed as "rapid genome engineering methodology", but it is time-consuming to insert the payload one by one. There may/should be more efficient way(s) to do it.*

It was our original intent, but attempts to target multiple landing pads have not been successful. We have added a discussion of this in the results.

6. *"there was a severe growth defect when grown in media lacking thiamine (Figure 1d)." > "Figure 1d" should be "Figure 1c"*

This typo has been corrected.

7. *"That was found to recover the transcription of the thi operon and overcome the growth defect (v2, Figure 1d)" > "v2, Figure 1d" should be "v2, Figure 1c".*

This typo has been corrected.

8. *"These sensors produce a 12- to 640-fold induction (Figure 2c, Appendix Figures 8 and 9) with low off states and no evidence of crosstalk (Figure 1c and Appendix Figure 7)." > "Figure 1c" should be "Figure 2c"*

This typo has been corrected.

9. *Appendix Figure 10b, the first 2 panels on the left column are the same.*

This typo has been corrected.

Reviewer #3:

1. *While I appreciate the technical advance demonstrated in the study, I have reservations about some of the sweeping criticisms the authors made about circuit engineering on plasmids. I am actually skeptical about some of them -- for instance, the growth impact of a circuit in a plasmid is not necessarily greater than that in the genome. It's just a matter of how the circuit is specifically engineered (e.g. plasmid copy number, circuit complexity, and choice of circuit components). Despite the best effort to insulate, an argument can be made that insertion of a large segment of DNA in the chromosome would always create an interruption (to various degrees) to the normal operation of the cell. My point is not to diminish what the authors have achieved. But I believe they should tone down the argument for advantages of integrating circuits into the chromosomes. What's better really depends on specific engineering objectives, which vary case by case. As noted by the authors, the major objective was "to simplify the process of genome engineering so that it approaches the ease of plasmid manipulations ...". This is an important and valuable objective that the authors have amply demonstrated.*

We have reduced material that could be perceived as overly critical of plasmids and have added some material to the discussion.

2. *I do not think the comparison between a circuit in a plasmid carrying a p15 origin (without selection) and one in the chromosome is fair. As noted by the authors, in the absence of antibiotic selection, many natural plasmids rely on other mechanisms to ensure stable maintenance. One prime example is the F plasmid, a ~100kb plasmid that can be stably maintained in standard lab conditions without positive selection. Again, I don't think the authors need to make this point to make the work valuable. If the authors do want to make a more general claim, as the writing appears to imply, a more rigorous experimental design is needed (where a circuit is optimized both on a plasmid and in the chromosome).*

We have added a direct comparison of a design where a circuit is optimized for a plasmid and in the chromosome (Figure 5). We have also added a discussion of the situations where plasmids are beneficial and methods to stabilize plasmids.

3. *When reporting the growth impact of different landing pads, the authors reported the OD values. It is not clear if what stage of the cultures these values correspond to. These values are useful to report; however, if the authors intend to make strong arguments based on the growth effects (or the lack of), it is more informative to show the growth rates of the cells carrying various landing pads. Two cultures with drastically different growth rates can reach the same OD when fully grown.*

The Methods have been clarified. Cells were grown for 5.5 hours in M9 media, corresponding to mid-exponential phase (OD₆₀₀ of 0.3 to 0.5). The measurements were taken prior to cells saturating.

4. *A major design strategy is the use of strong terminators to insulate. Because of this, it is useful to explain the quantification of these terminators in a greater depth in the results section. For example, T_s was defined in the methods section but it would be helpful to define it in the main text and to elaborate a bit more how it should be interpreted.*

We have added a detailed description of the definition and the interpretation of terminator strength in the main text. We have also added Appendix Figure 1 to clearly show the assay plasmid design and raw data used to calculate the terminator strength.

Thank you for sending us your revised manuscript. We think that the performed revisions address satisfactorily the issues raised by the reviewers. I am glad to inform you that your manuscript is now suitable for publication.

Before we formally accept the manuscript we would ask you to address a few remaining editorial issues.

Corresponding Author Name: Christopher A. Voigt

Manuscript Number: MSB-20-9584